# Noise as a Natural Regularizer in Markov Decision Processes: Connecting Environmental Stochasticity and Policy Simplicity

**Harry Chen** [1]   **Yiyang Sun** [2]   **Michal Moshkovitz** [3]   **Zachery Boner** [2]   **Lesia Semenova** [4]   **Cynthia Rudin** [2]
**Ronald Parr** [2]

## Abstract

The planning horizon in a Markov Decision Process (MDP) determines how far into the future an agent reasons. In practice, shorter horizons are commonly associated with policies that exhibit simpler or more interpretable decision-making behavior. In this paper, we establish a formal connection between environmental stochasticity and planning horizon in MDPs. We show that for broad classes of transition noise, solving a noisy MDP can be formally related to solving a noise-free MDP with a shorter effective discount factor, leading to identical optimal policies in some cases and near-optimal ones in others. We further characterize settings in which this correspondence breaks down, clarifying when horizon-based interpretations of noise are not valid. These results, which are supported by both theory and experiments, also give some insight into the common practice of using smaller discount factors for reinforcement learning than those that can be justified by standard modeling interpretations.

## 1. Introduction

A key concept in sequential decisions is the depth of the effective decision horizon. This is how far into the future an agent must look to make (approximately) optimal decisions. Shorter horizons offer many conveniences but also correspond to more myopic decisions. It is reasonable to expect that policies that are the product of shorter planning horizons will be easier to interpret or explain because fewer factors can influence particular choices of action. While we do not study representational interpretability directly, shorter dependencies can facilitate simpler policy structures in practice. We this as a useful property for interpretability in sequential decision-making, though it might not be sufficient on its own.

While a growing body of work studies interpretable policy representations in reinforcement learning (RL) (Verma et al., 2018; Paleja et al., 2023; Kearns et al., 1999; Trivedi et al., 2021), it largely focuses on model design rather than on how properties of the environment contribute to interpretability. In supervised learning, noise has been shown to enlarge the set of near-optimal models, enabling simpler hypotheses to perform well (Semenova et al., 2022; 2023; Boner et al., 2024). In RL, however, uncertainty propagates over time through the transition dynamics. In this paper, we demonstrate that environmental stochasticity can play an analogous role by limiting the planning horizon.

In RL, the depth of planning is tied directly to the choice of the discount factor (Sutton & Barto, 1998; Kearns et al., 1999). In practice, however, the chosen discount is often smaller than what would be naturally implied by common modeling rationales such as inflation, termination probability, or probability of catastrophic failure. Empirically, smaller discount factors often stabilize learning and are associated with more robust behavior (Hu et al., 2022; Amit et al., 2020). Perhaps there is a more compelling explanation for this than mere convenience or rules of thumb.

In this work, we address this question by studying how uncertainty in the transition dynamics shapes the effective planning horizon. For several fundamental noise models, we show that *transition noise can act as a structural factor that constrains the effective planning problem*, admitting exact or approximate reductions to noise-free MDPs with modified discount factors. We prove that these effects are mathematically and empirically justified and summarize the results in Table 1. Specifically, we ask when a noisy MDP induces the same or similar optimal behavior as a noise-free MDP under reduced discounting.

Using this perspective, we establish exact equivalence results for two practically relevant classes of transition noise. For "sticky noise,", where actions may fail and leave the

[1]Massachusetts Institute of Technology, Cambridge, MA, USA
[2]Duke University, Durham, NC, USA [3]Google Research, Israel
[4]Rutgers University, New Brunswick, NJ, USA. Correspondence to: Harry Chen <harryc60 AT mit.edu>.

*Proceedings of the 43rd International Conference on Machine Learning*, Seoul, South Korea. PMLR 306, 2026. Copyright 2026 by the author(s).

*Table 1.* **Summary of transition noise models and their relationship to discounting.** Here $\gamma$ denotes the original discount factor of the MDP, and $\gamma'$ denotes the effective discount factor induced by transition noise. $\beta$ denotes the probability of action failure in sticky noise, and $\alpha$ denotes the probability of remaining in the original dynamics under reset noise.

| Noise model | Effective discount | Relationship |
|---|---|---|
| Sticky noise | $\gamma' = \dfrac{\beta\gamma}{1 - \gamma(1 - \beta)}$ | Exact equivalence: Planning is equivalent to a shorter-horizon MDP |
| Reset noise | $\gamma' = \alpha\gamma$ | Exact equivalence: Planning is equivalent to a shorter-horizon MDP |
| Reversible and slowly changing noise | Reduced discount $\gamma' < \gamma$ used to derive near-optimal policies | Near-optimality guarantee: Short-horizon policies remain near-optimal |
| General transition noise | No useful discount-equivalent representation | Discounting alone is insufficient: Weak bounds from generic smoothness assumptions, or no bounds at all |

agent in its current state, and for reset-style noise, where the agent is intermittently returned to a baseline state distribution, we show that solving the noisy MDP is *exactly* equivalent to solving a noise-free MDP with a reduced discount factor. In both cases, this equivalence results in the same optimal policy under a reduced planning horizon (Sections 4–5). Together, these results demonstrate that for these practically relevant noise models, environmental stochasticity can be precisely captured through discounting without loss of optimality.

We then extend our analysis beyond exact equivalence to increasingly general classes of transition noise. For reversible (noise effects can be undone by a single action), slowly changing (reward differences between noise-connected states are bounded) noise models, we show that policies optimized under reduced discounting remain near-optimal in the corresponding noisy MDPs, effectively limiting the depth of temporal reasoning required (Section 6). For fully general noise, we discuss worst-case bounds relating noisy MDPs to discounted noise-free MDPs, clarifying both the reach and the limitations of discount-based approximations (Section 7). We also present counterexamples demonstrating that not all noise admits a discount-equivalent representation, highlighting the importance of noise structure rather than noise magnitude alone. Complementary empirical results in Section 8 illustrate these phenomena.

Our results also clarify how our perspective differs from robustness-based approaches to uncertainty in reinforcement learning (Iyengar, 2005; Nilim & El Ghaoui, 2005; Wiesemann et al., 2013). Robust MDP methods typically model uncertainty through adversarial or worst-case transition perturbations, yielding policies that are optimized to perform well under such extremes. Our work studies a different regime of uncertainty, one that is common in practice: stochastic but non-adversarial transition noise. From this perspective, transition noise is not an obstacle to be defeated, but a structural property of the environment that can naturally constrain long-term planning.

Taken together, we provide an explanation for when and why environmental uncertainty leads to horizon-limited planning, offering a principled connection between stochasticity, dis-

counting, and the emergence of simpler, more interpretable decision-making behavior in reinforcement learning. From an interpretability perspective, our results suggest that simpler, i.e. shorter horizon planning, may emerge naturally as a consequence of environmental uncertainty.

## 2. Related Work

**Discount factor and noise.** Beyond encoding time preference, the discount factor has long been used as a form of regularization in reinforcement learning. Prior work shows that reducing $\gamma$ in approximate dynamic programming can decrease errors introduced by function approximation and improve solution quality (Petrik & Scherrer, 2008), and that shorter planning horizons can mitigate overfitting to model errors (Jiang et al., 2015). Complementary approaches achieve robustness by increasing environmental stochasticity, such as domain randomization in sim-to-real transfer for robotics (Peng et al., 2018). Our work formally unifies these perspectives. We show that, for broad classes of transition noise, solving a noisy MDP is equivalent to solving a noise-free MDP with a reduced discount factor, establishing transition noise as a mechanism for shortening the effective planning horizon.

**Robust MDPs.** A substantial literature addresses model uncertainty via Robust Markov Decision Processes (MDPs), where transition dynamics are assumed to lie in a predefined uncertainty set, and an adversary selects the realization that minimizes expected, discounted return (Nilim & El Ghaoui, 2005; Givan et al., 2000; Iyengar, 2005; Goyal & Grand-Clément, 2018). This framework is inherently worst-case, aiming to guarantee performance under adversarial perturbations. Our approach differs in both scope and objective. Rather than developing planning algorithms robust to broad uncertainty sets, we focus on specific noise models to extract qualitative insights. In particular, we analyze the structural relationship between transition noise and the discount factor $\gamma$, rather than proposing a new, robust planning method.

**Simplicity in supervised learning and policy representation.** Noise can regularize supervised learning problems by enlarging the set of near-optimal hypotheses, making

simpler models statistically indistinguishable from more complex ones (Semenova et al., 2023; Boner et al., 2024; Bishop, 1995; Paes et al., 2023). These results characterize simplicity in static prediction settings and do not address how noise affects temporal structure or planning in sequential decision-making. Our work studies this gap by analyzing how transition stochasticity shapes the structure of optimal policies in MDPs. In parallel, much of the literature on interpretable reinforcement learning enforces transparency through constrained policy representations, including programmatic or symbolic policies (Verma et al., 2018; Paleja et al., 2023), decision tree or rule-based policies (Silva et al., 2020; Bastani et al., 2018), or prototype-based policies (Kenny et al., 2023). By contrast, our work does not restrict representations, but studies how properties of the environment, specifically transition stochasticity, shape the effective planning horizon.

## 3. Preliminaries and Notation

We consider a Markov Decision Process (MDP) $M = (S, A, P, R)$. The state space $S$ is finite with $|S| = n$, and the action space $A$ is discrete. The transition model $P$ is given by $P(s'|s, a)$, where $s, s' \in S$, $a \in A$. The reward function $R : S \to [0, 1]$ is defined over states. We evaluate this MDP using various discount factors $\gamma \in (0, 1)$. A policy $\pi : S \to A$ induces a transition matrix $P^\pi$. The value function $V_\gamma^\pi \in \mathbb{R}^n$ satisfies

$$V_\gamma^\pi = R + \gamma P^\pi V_\gamma^\pi = (I - \gamma P^\pi)^{-1} R = \sum_{i=0}^\infty \gamma^i (P^\pi)^i R.$$

A policy $\pi^*$ is optimal if $V_\gamma^{\pi^*} \geq V_\gamma^\pi$ for all policies $\pi$ (componentwise). To facilitate our analysis of the interplay between transition noise and the planning horizon, we distinguish between three distinct value functions:

- $V_\gamma^\pi$: The value function in a *noise-free, clean environment*. This will be evaluated using both the original discount factor $\gamma$ and an alternative $\gamma' < \gamma$ (denoted $V_{\gamma'}^\pi$) to represent a shorter effective horizon;

- $V_{\gamma,\text{noisy}}^\pi$: The value under the original discount $\gamma$, but calculated within the noisy MDP.

For a discounted, infinite horizon MDP, the planning horizon is nominally infinite, but discounting attenuates the impact of future rewards. For any discount factor and any tolerable-error $\epsilon$, one can compute a horizon length $\tau$ beyond which future choices can be ignored with penalty of no more then $\epsilon$ on the quality of decisions made. This so-called $\epsilon$-horizon (Kearns et al., 1999), $\tau$, can be used to determine a maximum number of steps of value iteration needed, or the depth of a search needed to determine (or possibly explain) the decision taken from a single state.

The discount factor, $\gamma$, in MDPs is often viewed as a measure of future uncertainty. One interpretation of the discount factor is that it represents a $1 - \gamma$ probability of death (transition to a state with value 0) at every time step. Anecdotally, researchers have likened other types of uncertainty to discounting but, to our knowledge, this connection has not been formalized to the extent done in this paper.

## 4. Sticky Noise

We begin our analysis with sticky noise, a model where actions have a constant probability of failing and leaving the agent in its current state, effectively inducing a self-loop in the transition dynamics.[1] $\beta \in [0, 1]$ determines a tendency to stay in the same state with probability $1 - \beta$ for *any* $\pi$:

$$P_\beta^\pi = \beta P^\pi + (1 - \beta)I,$$

where $I$ is an identity matrix. When $\beta = 1$ we have our original $M$, and when $\beta = 0$, it is impossible to change state. One motivation for sticky noise is actions that have a small probability of failing at any time. Another motivation is an artifact of using a discretized model of a continuous environment: If an agent's position in the continuous environment is in a corner of a cell of the discretized model, and executes an action designed to move to an adjacent cell, that action outcome may not always reach the target cell when mapped back to the discretized model; the agent may move to an opposing edge or vertex of the current cell, rather than the next cell. This is often modeled as sticky noise.

The next theorem shows that under sticky noise, solving the noisy infinite-horizon MDP is equivalent to solving a noise-free MDP with a smaller discount factor, yielding the same optimal policy. The proof is in Appendix A.1.

**Theorem 4.1** (Sticky Noise). *For an MDP $M = (S, A, P, R)$ and its sticky counterpart $M_\beta$ with sticky parameter $\beta \in [0, 1]$, let $\pi^*$ be the optimal policy for $M_\beta$ under a discount factor $\gamma \in (0, 1)$. Then $\pi^*$ is also the optimal policy for the original MDP $M$ under a modified discount factor $\gamma' = \frac{\beta\gamma}{1 - \gamma(1 - \beta)} < \gamma$.*

Real environments, however, may exhibit more structured noise: action failures often produce state- and action-dependent stochastic mixtures rather than pure self-loops. Nevertheless, the horizon-shortening effect of sticky noise can still apply if the noise contains a minimal sticky component. One example of how sticky components may arise is through two general assumptions. Suppose that we have an original MDP and a noisy MDP with transition probabilities $P$ and $P_\text{noisy}$ respectively. The *controllable self transition* assumption means that, in $P$, the agent can deterministi-

---

[1]This is distinct from sticky actions, as sometimes implemented in, e.g., Atari games. Sticky actions are a non-Markov form of noise where the intended action is applied for multiple time steps.

cally stay in the current state, but it cannot inadvertently do so. That is, for all states $s \in S$, there exists some action $a \in A$ such that $P(s|s,a) = 1$ and $P(s|s,a') = 0$ for all $a' \in A \setminus \{a\}$. The *convex noise* assumption states that noisy MDP actions are mixtures of actions in the original MDP, i.e., for all state-action pairs $(s,a)$:

$$\exists k, \exists c_1, \ldots, c_k : P_{\text{noisy}}(s'|s,a) = \sum_{i=1}^{k} c_i P(s'|s,a_i), \quad (1)$$

where all $c_i \geq 0$, and $\sum_{i=1}^{k} c_i = 1$. Note that the quantification above allows the mixture to vary at different states. We do require, however, that all actions in all states exhibit a minimum level of stickiness: $\forall s, a : P_{\text{noisy}}(s|s,a) \geq 1 - \beta$. To get a lower bound on the value function, we consider a modification to $P_{\text{noisy}}$ called $P_{-\text{noisy}}$, which removes the common sticky component from $P_{\text{noisy}}$:

- $\forall s, a: P_{-\text{noisy}}(s|s,a) = \frac{1}{\beta}(P_{\text{noisy}}(s|s,a) - (1-\beta))$,

- $\forall s, a, s' \neq s : P_{-\text{noisy}}(s'|s,a) = \frac{P_{\text{noisy}}(s'|s,a)}{\beta}$.

In general, we will use $V_{\gamma,P}^{\pi}$ to be the value of policy $\pi$ for the MDP with transition function $P$ and discount $\gamma$. We show that the optimal policy of the noisy MDP is sandwiched between policies that are optimal for MDPs with reduced discount factors.

**Theorem 4.2** (Generalized Sticky Noise). *Consider $P_{\text{noisy}}$ and $P_{-\text{noisy}}$ as defined above, $P_\beta$ the original transition model with sticky noise $\beta$ added, and $\gamma' = \frac{\beta\gamma}{1-\gamma(1-\beta)}$. Define the following policies: $\pi_\gamma^*$ is optimal for the original MDP, $\pi_{\gamma'}^*$ is optimal for the original MDP with discount $\gamma'$, $\pi_{\gamma,\text{noisy}}^*$ is optimal for $P_{\text{noisy}}$ and $\gamma$, $\pi_{\gamma',-\text{noisy}}^*$ is optimal for $P_{-\text{noisy}}$ and $\gamma'$. Then we have:*

$$V_{\gamma,\text{noisy}}^{\pi_{\gamma',-\text{noisy}}^*} = V_{\gamma,\text{noisy}}^{\pi_{\gamma,\text{noisy}}^*} \leq V_{\gamma,\beta}^{\pi_{\gamma'}^*} \leq V_{\gamma}^{\pi_\gamma^*}.$$

If the controllable self transition and convex noise assumptions do not hold, but the minimum stickiness assumption is otherwise achieved, then the equality in the theorem still holds, but the upper bounds do not apply. The proof is in Appendix A.1. This theorem supports a recurring theme: Noise shortens the planning horizon.

## 5. Reset Noise

For all states and actions, suppose there is a fixed noise distribution $\mu$ that is mixed with each state's next state distribution. One way to think of this is as some sort of reset noise, by which with some probability $1 - \alpha$, the agent is reset to some distribution over states. This probability and the reset state distribution are independent of the agent's current state and action. This can be thought of as a probability

of the robot getting "kidnapped" by a human and moved to another location, or a game being reset to a starting configuration. Other realizations of this could include a random outcome in a game (such as going to jail in Monopoly™ or the bat in Hunt the Wumpus) that can happen from any state. We note that sticky noise is not a special case of reset noise since sticky noise is state-dependent.

Following the convention of the previous section, we define the MDP $M_\alpha$ under reset noise, where $\alpha$ determines the amount of reset noise that is added to the transition matrix for every policy. Define $P^\mu$ to be an $n \times n$ matrix with $\mu$ in each row. For any $\pi$ and any $\alpha$, we have: $P_\alpha^\pi = \alpha P^\pi + (1-\alpha)P^\mu$. Thus, when $\alpha = 1$ we have our original $M$, and when $\alpha = 0$, all states transition to the reset distribution $\mu$ under any policy. Then we have:

**Theorem 5.1** (Reset Noise). *For an MDP $M = (S, A, P, R)$ and its reset counterpart $M_\alpha$, let $\pi^*$ be the optimal policy for $M_\alpha$ under a discount factor $\gamma \in (0,1)$. Then $\pi^*$ is also the optimal policy for the original MDP $M$ under discount factor $\gamma' = \alpha\gamma < \gamma$.*

Theorem 5.1 (see proof in Appendix A.1) shows that solving an MDP $M_\alpha$, which adds reset noise parameterized by $\alpha$ to MDP $M$ is equivalent to solving another MDP $M'$ which has the same transition model $T$ as $M$, but has the discount factor reduced from $\gamma$ to $\gamma\alpha$. Similar to sticky noise, the reset noise results can in principle be generalized to cases where there is a shared reset component that is part of a more general noise model.

A remarkable aspect of Theorem 5.1 is that the result is independent of the reset noise distribution. This observation offers insight into the common practice of using smaller discount factors in training than would be directly justified by explicit modeling assumptions. In particular, if a practitioner believes that there is substantial reset noise that is not accurately modeled, choosing a smaller discount factor can be viewed as a principled way to hedge against this uncertainty without explicitly modeling the noise itself.

Since sticky noise and reset noise each result in an MDP with the same optimal policy but a smaller discount factor, the analysis can be combined sequentially to model MDPs that contain both types of noise. Next, we study whether similar horizon-limiting effects persist under more general, reversible noise models. Because these noise models are structurally more complex, exact equivalence to a discounted noise-free MDP can no longer be guaranteed. Instead, we focus on establishing bounds and near-optimality guarantees.

## 6. Slowly-Changing Reversible MDPs

Sections 4 and 5 establish exact equivalences between structured transition noise and reduced discounting. These re-

sults, however, greatly rely upon the noise structure. In this section, we move beyond exact equivalence and show that, under *reversibility* and *slowly-changing reward* assumptions (that we define next), policies optimized with reduced discounting remain near-optimal in the noisy MDP with only slight modification.

We consider a noise model that interpolates between the original dynamics $P$ and a noise transition function $P'$. For a fixed parameter $\epsilon \in (0, \frac{1}{2})$, we define the noisy MDP $M_\epsilon$ with transition function:

$$P_{\text{noisy}}(s'|s,a) := (1-\epsilon)P(s'|s,a) + \epsilon P'(s'|s,a)$$

This formulation treats actions as following the original MDP with a probability $1-\epsilon$, while a probability $\epsilon$ exists for the *noise* dynamics $P'$ to activate and override the intended transition.

We define a noisy MDP $M_\epsilon$ to be reversible[2] if for every pair of states $(s, s')$ where $\max_{a \in A} P'(s'|s,a) > 0$, there exists an action such that $\max_{a \in A} P(s|s',a) = 1$. This property ensures that the effects of noise can be reversed deterministically through a single action. Furthermore, we say $M_\epsilon$ is $(C_{\min}, C_{\max})$-slowly-changing (for $C_{\min}, C_{\max} \geq 0$) if the reward difference between such states satisfies $-C_{\min} \leq R(s') - R(s) \leq C_{\max}$. This is more strongly satisfied in cases where rewards are dense.

Together, a reversible, $(C_{\min}, C_{\max})$-slowly-changing noise model avoids two primary ways that noise can catastrophically change the behavior of the optimal policy. Reversibility prevents noise from sending the agent to "traps" from which the agent cannot return, and the slowly-changing assumption prevents single-occurrences of noise from incurring unbounded penalty.

The following theorem shows how the optimal policy for a more heavily discounted MDP can still be a good policy for the noisy MDP:

**Theorem 6.1.** *For MDP $M = (S, A, P, R)$ and a $(C_{\min}, C_{\max})$-slowly-changing reversible noisy MDP $M_\epsilon = (S, A, P_{\text{noisy}}, R)$, define $\gamma' < \gamma$ to be $\gamma' = \frac{2\gamma(1-\epsilon)}{1+\sqrt{1-4\gamma^2\epsilon(1-\epsilon)}}$. Let $\pi_{\gamma'}^*$, $\pi_\gamma^*$, and $\pi_{\gamma,\text{noisy}}^*$ maximize $V_{\gamma'}^\pi$, $V_\gamma^\pi$, and $V_{\gamma,\text{noisy}}^\pi$ respectively. If $V_{\gamma,\text{noisy}}^{\pi_{\gamma,\text{noisy}}^*} \leq V_\gamma^{\pi_\gamma^*}$, then there exists a nonstationary policy $\pi'$ obtained from $\pi_{\gamma'}^*$ such that*

$$V_{\gamma,\text{noisy}}^{\pi'} \geq \frac{1-\gamma'}{1-\gamma} V_{\gamma'}^{\pi_{\gamma'}^*} - \frac{C_{\min}\epsilon\gamma}{1-2\epsilon} \cdot \frac{1}{1-\gamma}$$

$$\geq V_{\gamma,\text{noisy}}^{\pi_{\gamma,\text{noisy}}^*} - \left( \frac{C_{\min}\epsilon\gamma}{1-2\epsilon} + 2\epsilon(1-2\epsilon)^{\frac{1-2\epsilon}{2\epsilon}} \right) \cdot \frac{1}{1-\gamma}.$$

---

[2]This property can be relaxed so that the probability of reversing is less than 1, although the set of pairs of states considered must be expanded to include unintended states visited when taking reversing actions.

Theorem 6.1 bounds the potential loss in performance on the noisy MDP when utilizing a policy optimized on the clean MDP with smaller discount factor $\gamma'$, which decreases as $\epsilon$ increases. Unlike for pure sticky or reset noise, we guarantee that the value functions for $\pi'$ and $\pi_{\gamma,\text{noisy}}^*$ are similar, but not that the policies themselves are.

The assumption that $V_{\gamma,\text{noisy}}^{\pi_{\gamma,\text{noisy}}^*} \leq V_\gamma^{\pi_\gamma^*}$ is reasonable since, for a fixed discount factor, noise typically makes the MDP harder for an agent. In particular, if for every $s \in S$ and $a' \in A$, $P_{\text{noisy}}(\cdot|s,a')$ is a convex combination of the distributions $\{P(\cdot|s,a) : a \in A\}$, then $V_{\gamma,\text{noisy}}^{\pi_{\gamma,\text{noisy}}^*} \leq V_\gamma^{\pi_\gamma^*}$ holds.

At a high level, $\pi'$ acts exactly as $\pi_{\gamma'}^*$ does until noise activates, after which $\pi'$ reverses the effects of noise and resumes behaving like $\pi_{\gamma'}^*$. From a theoretical perspective, $\pi'$ is only slightly more complex than $\pi_{\gamma'}^*$, and the noise-reversal is very short-term behavior, implying that noisy environments permit simpler, smaller-horizon, near-optimal policies. From a practical perspective, if $\pi'$ can be implemented, then Theorem 6.1 shows that a practitioner can achieve near-optimal performance on a noisy environment by simply fitting on a clean environment with a reduced discount factor and then deploying $\pi'$. This provides the benefit of faster convergence rates and greater robustness to potentially unknown noise distributions. If the noisy transition function is known, then $\pi'$ is immediately implementable without refitting. Otherwise, the user can either estimate the noisy transition function through observation, or the user can keep track of Q-values after deployment and take the action with the highest Q-value.

We highlight that, in realistic or continuous MDPs, either reversibility or the slowly-changing assumption may not hold exactly and so Theorem 6.1 cannot be applied. In these settings, approximate versions of these assumptions hold when the agent retains some level of ability to control for noise, or when the reward does not change greatly between most but not all states. We therefore expect that the general principle of Theorem 6.1, that the effects of noise can be converted into a time cost and therefore an effective reduction in the discount factor, will still hold broadly. We support this in Section 8.2 through our experiments with the Reacher environment.

### 6.1. Proof Outline for Theorem 6.1

Theorem 6.1 follows immediately from the next two lemmas. Lemma 6.2 formalizes the effect of reversible, slowly-changing noise by constructing a nonstationary policy that "undoes" noise events and bounding the resulting deviation in value. Lemma 6.3 then provides a generic bound relating value functions under discount factors $\gamma$ and $\gamma'$, independent of the noise model.

**Lemma 6.2.** *In the setting of Theorem 6.1, for every policy*

$\pi$ on $M$, there exists a modified nonstationary policy $\pi'$ on $M_\epsilon$ satisfying

$$-\frac{C_{\min}\epsilon\gamma}{1-2\epsilon} \leq (1-\gamma)V_{\gamma,\text{noisy}}^{\pi'} - (1-\gamma')V_{\gamma'}^{\pi} \leq \frac{C_{\max}\epsilon\gamma}{1-2\epsilon}.$$

**Lemma 6.3.** *In the setting of Theorem 6.1, for any policy $\pi$, for all $s \in S$:*

$$V_\gamma^\pi(s) \leq \frac{1-\gamma'}{1-\gamma}V_{\gamma'}^\pi(s) + 2\epsilon(1-\epsilon)^{\frac{1-2\epsilon}{2\epsilon}} \cdot \frac{1}{1-\gamma}.$$

Lemma 6.3 can be shown by bounding the difference between two geometrically-weighted sequences of rewards and then applying the definition of $\gamma'$. We therefore end this section by presenting a high-level outline of the proof for Lemma 6.2 (See Appendix A for full details). We additionally show through this proof that our bound is tight.

**Proof Outline for Lemma 6.2**:

1. **Construction of $\pi'$:** We construct a modified policy $\pi'$ that mimics $\pi$ by reversing the effects of noise. Specifically, $\pi'$ follows the actions prescribed by $\pi$ until noise activates when transitioning out of a state $s$. It then repeatedly takes actions to return to $s$ until successful, after which it resumes following $\pi$. Since noise may activate consecutively, returning to $s$ may take multiple timesteps. Although $\pi'$ assumes knowledge of when noise occurs, it can be replaced with any nonstationary policy whose trajectories follow the same distribution as $\pi'$ without needing knowledge of when noise occurs, and we often only care about stationary policies actually followed by the agent whose values would then be theoretically bounded below by those of $\pi'$.

2. **Reward Coupling:** For each fixed trajectory of $\pi$ on the original MDP, we couple a corresponding noisy trajectory generated by $\pi'$ in the noisy MDP. We express the expected reward of this noisy trajectory as the reward of the fixed trajectory plus an additional term capturing changes due to visits to unintended states caused by noise.

3. **Bounding Changes in Reward:** Finally, we bound these reward changes using the slowly-changing property. The resulting bound is independent of the underlying MDP and noise model, allowing us to construct a worst-case noise model and compute the bound tightly in closed form.

## 7. Limits of the Noise-Discount Equivalence

Our results so far have relied upon specific noise models to establish a meaningful relationship between noise and horizon shortening. What can be said if there are extremely weak or no assumptions about the form of the noise?

### 7.1. Weak Assumptions On Noise Give Weak Bounds

First, we describe the high level intuition for how noise that the blurs distinction between actions can shorten the planning horizon: If all actions tend to have the same distribution over next states as noise increases, then there is less benefit to planning since different policies will tend to produce increasingly similar distributions over future states.

Prior results from Jiang et al. (2016), which were motivated somewhat differently, can nevertheless help give insight into this effect. Their Theorem 2 bounds the loss from using the optimal policy for one discount factor on an MDP with a different discount factor. Their bound is increasing in two terms, $\delta_P$ and $\kappa_\gamma$, where

$$\delta_P = \max_{s \in S, a, a' \in A} \|P(\cdot|s,a) - P(\cdot|s,a')\|_1,$$

is the maximum 1-norm difference in next state distributions between different actions at the same state, and

$$\kappa_\gamma = \max_{s, s' \in S} |V_\gamma^{\pi^*}(s) - V_\gamma^{\pi^*}(s')|$$

is the maximum difference in value between any two states under the optimal policy. Their bound is:

$$\|V_\gamma^{\pi_\gamma^*} - V_\gamma^{\pi_{\gamma'}^*}\|_\infty \frac{\frac{\delta_P}{2}\kappa_\gamma(\gamma - \gamma')}{(1-\gamma)(1-\gamma(1-\frac{\delta_P}{2}))}.$$

If actions start off having distinct deterministic outcomes, then $\delta_P$ will be at its maximal value of 2. Adding small amounts of noise would still result in large values of $\delta_P$, e.g., sticky noisy of 0.1 could result in $\delta_P = 1.8$. When $\delta_P$ is close to 2, the denominator in the bound will approach $(1-\gamma)$. Thus, $\delta_P$ will start to get small only when there is so much noise that all actions have similar outcomes.

It is possible for $\kappa_\gamma$ to be arbitrarily close to $\frac{1}{1-\gamma}$ in the original MDP for certain transition models, so when the added noise is low, the contribution of this component can scale with $\frac{1}{1-\gamma}$. Uniform reset noise can force this to be as low as 1 as the reset noise approaches 100%. Thus, if noise blurs the distinctions between actions, then it will tend to decrease $\delta_P$ and $\kappa_\gamma$, reducing the suboptimality.

To underscore the importance of studying specific noise models, we emphasize that our approach allows us to achieve exact equivalence (0 optimality loss) for any value of sticky noise and reset noise. In the slowly changing, reversible case, the optimality loss is minimized when the added noise approaches 0. Adapting generic bounds like those from Jiang et al. (2016) primarily gives insight into limiting behavior when the effect of noise is large enough that different actions tend to have nearly the same effects, whereas our results provide strong bounds even when the level of noise is low.

## 7.2. When Noise and Discounting Do Not Align At All

We demonstrate that in environments containing catastrophic risks, the optimal policy under noise can be structurally distinct from any policy derived from the noiseless setting, regardless of the discount factor used. Intuition. Consider a "Tightrope Walk" scenario (Figure 1). The agent must choose between crossing a dangerous tightrope to reach a massive treasure or taking a safe walking path to a small reward. In a deterministic world (noiseless), the tightrope is perfectly safe; thus, no matter how shortsighted (small $\gamma$) or longsighted (large $\gamma$) the agent is, it will always prefer the massive treasure. However, in the real world (noisy), the tightrope carries a risk of falling to a catastrophic state. If this risk is high enough, the optimal behavior flips to taking the safe path. Since the "safe path" policy is never optimal in the noiseless setting (regardless of discounting), discounting fails to act as a proxy for this type of noise. We provide the formal theorem and proof of this "Tightrope" counterexample in Appendix A.3.

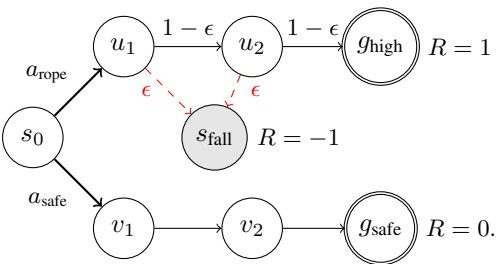

*Figure 1.* **The Tightrope MDP.** The agent chooses between a risky "tightrope" path (top) with high reward but a risk of catastrophic failure (red dashed arrows), and a safe path (bottom) with low reward. Both paths consist of $k$ steps (drawn here with $k = 2$). In the noiseless case ($\epsilon = 0$), the top path is preferred for any $\gamma$. In the noisy case, the preference flips to the safe path.

In the next section, we complement our theoretical analysis with empirical studies designed to illustrate these phenomena in practical reinforcement learning settings. While the theoretical results do not depend on specific algorithmic choices, the experiments provide intuition for how different noise structures and discount choices manifest in learned policies and planning behavior.

## 8. Experimental Results

Although our theoretical results provide exact or approximate equivalences between noise and discounting for sticky, reset, and slowly-changing reversible noise, it is often the case that realistic transition noise models do not fall cleanly into any of these categories. Our experiments are designed to investigate and verify when this equivalence holds and when it breaks down. To do this, we explore two different environments: a small, discrete grid environment and the more complex, continuous reacher environment. We con-

sider both reset noise and a broader uniform action noise model, showing that uniform noise often mimics reduced discounting, but fails when it can induce highly penalized or irreversible transitions (see Appendix B).

### 8.1. Grid Environment

We begin by examining a classical grid environment – a discrete environment with a limited number of states, which permits the use of dynamic programming-based algorithms to exactly compute the optimal policy. In this environment, an agent must learn to navigate within a 2-dimensional grid map. Each square in the map is either empty or a wall, where the set of states corresponds to the set of empty squares, and walls are impassable. At every timestep, the agent has five actions available, each of which acts as a deterministic state transition: Move exactly one square left, right, up, down, or stay still. Rewards are state-based, with each empty square containing an integer reward.

We explore two noise models, uniform noise and reset noise. For uniform noise with parameter $\epsilon$, there is an $\epsilon$ probability at each timestep for noise to activate, causing the agent to move in one of the four cardinal directions uniformly and at random. This can be interpreted as slippage, causing the trajectory to change. For reset noise with parameter $\alpha$, there is a $1 - \alpha$ probability at each timestep for noise to activate, causing the agent to transition to the reset distribution.

For general maps, we may expect a failure of equivalence between the discount factor and noise whenever noise can sharply and immediately incur large penalties in reward. We therefore focus on the hallway map, where noise does not catastrophically affect the agent. As shown in Figure 2, there are two rows in this map. The top row contains rewards in increasing order from $+1$ to $+9$ separated by walls, and the bottom row contains squares with no reward but is otherwise fully traversable. Because optimal policies exhibit simple threshold structure, this environment provides a clear setting to compare the effects of noise and discounting. We analyze how these effects manifest in the optimal policies for this MDP, with an additional grid-based example deferred to Appendix B.

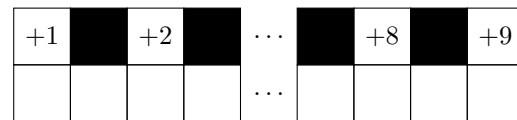

*Figure 2.* **The hallway MDP.** White squares are traversable states, black squares are impassable, and squares with nonzero reward are labeled with the reward value.

To compare the effects of noise and of lowering the discount factor, we provide a quantitative measure of how distant two policies are. For a policy $\pi_\gamma^*$ optimizing the value function $V_\gamma^\pi$, we de-

fine $A^*(\pi_\gamma^*, s) = \arg\max_{a \in A} \sum_{s' \in S} P(s'|s, a) V_\gamma^{\pi_\gamma^*}(s')$ to be the set of actions at state $s$ that perform equally optimally. Similarly, for a policy $\pi_{\gamma,\text{noisy}}^*$ optimizing $V_{\gamma,\text{noisy}}^\pi$, we define $A_{\text{noisy}}^*(\pi_{\gamma,\text{noisy}}^*, s) = \arg\max_{a \in A} \sum_{s' \in S} P_{\text{noisy}}(s'|s, a) V_{\gamma,\text{noisy}}^{\pi_{\gamma,\text{noisy}}^*}(s')$. We measure the distance between $\pi_{\gamma_1}^*$ and $\pi_{\gamma_2,\text{noisy}}^*$ to be the percentage of states $s$ for which $A^*(\pi_{\gamma_1}^*, s) \cap A_{\text{noisy}}^*(\pi_{\gamma_2,\text{noisy}}^*, s) = \emptyset$. This metric directly captures whether noise-induced behavior can be reproduced by discount reduction.

Using this metric, we measure for which pairs of discount factors the optimal policies for the clean and noisy environment for coincide. On the clean MDP, the uniform noise MDP with $\epsilon = 0.3$, and the reset noise MDP with $\alpha = 0.95$ and whose reset distribution is uniform over the bottom row, we compute optimal policies as $\gamma$ is varied in the interval $[0.70, 0.99]$. We then compare every pair of policies between the clean and noisy MDPs, which we show in Figure 3. As we can see, optimal policies are most similar when the discount factor for the clean MDP is smaller than the discount factor for the noisy MDP, matching our theoretical results. Furthermore, for reset noise, we see an exact equivalence as predicted by Theorem 5.1. We also observe an exact equivalence for sticky noise, which we elaborate on in Appendix B.1.

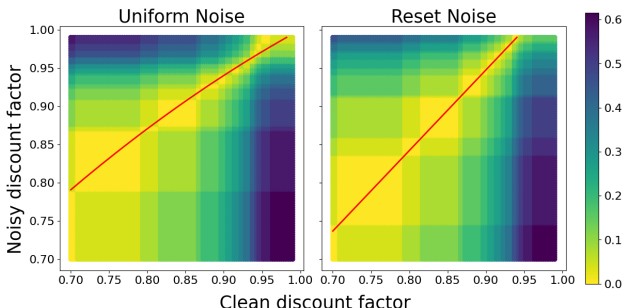

*Figure 3.* The distance between optimal policies on the clean hallway MDP and the noisy hallway MDP with uniform noise with $\epsilon = 0.3$ and reset noise with $\alpha = 0.95$. The red lines depict the $(\gamma, \gamma')$ pairs predicted by Theorems 6.1 and 5.1 respectively. For uniform noise, we use an effective probability of noise of $0.225$ to compute $\gamma'$ since there is a $25\%$ chance that uniform noise takes the action the agent intended to take.

## 8.2. Reacher Environment

To evaluate the robustness of policy learning under continuous control, we use a 2-Link Reacher environment developed within the Gymnasium (Towers et al., 2025) and MuJoCo (Todorov et al., 2012) frameworks[3]. The agent controls a planar robotic arm with hinge joints, tasked with

---

[3]All code for the experiments on the Reacher environment can be found at https://github.com/Extile1/MDP_Noise_Regularization.

reaching a stochastically spawned target $g \in \mathbb{R}^2$ within a confined workspace. To test stability and generalization, we introduced specific modifications to the standard setup.

First, we employ a dense reward function that heavily penalizes actions. In order to make the optimal policy more sensitive to the discount factor, we increased the control weight $w_{\text{ctrl}}$ of the action $a_t$ from **1.0** to **5.0** so the reward function penalizes large actions to a greater degree:

$$r_t = -\|\mathbf{p}_{\text{tip}} - g\|_2 - 5.0\|a_t\|_2^2. \tag{2}$$

where $\mathbf{p}_{\text{tip}}$ is the position of the tip of the 2-link reacher. Secondly, to simulate hardware instability and assess policy recovery, we implement a random action noise. With probability $p_{\text{noise}}$, the intended action is replaced by a uniform random value. This serves as an aggressive entropy injection, making the trajectory less smoothly controllable.

Policies were trained using Proximal Policy Optimization (PPO) (Schulman et al., 2017). Hyperparameters were optimized via Optuna (Akiba et al., 2019). A detailed experimental setup is in Appendix C.

### 8.2.1. DISCOUNT BEHAVIOR WITHIN REACHER

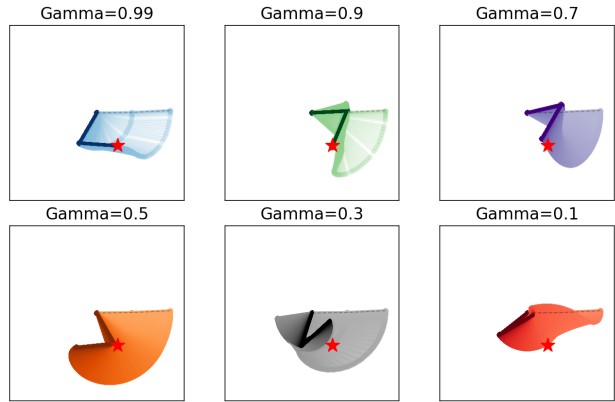

*Figure 4.* An example trajectory changes as the discount factor changes. The star represents the target. Darker colors represent states at later timestamps.

Figure 4 shows different behaviors of reacher under different discount factors: At low discount factors, the agent acts greedily, often stopping short or drifting toward the workspace center to minimize control effort. When the discount factor starts to increase ($\gamma \in [0.7, 0.9]$), it allows the agent to reach the target, but the arm "wrangles" around the goal rather than staying still. Conversely, $\gamma = 0.99$ yields stable convergence, achieving high precision and a steady, motionless hold.

### 8.2.2. RESULTS WITHIN RANDOM ACTION NOISE

Here, we investigate the effect of random action noise on policy stability by varying the noise probability $p_{\text{noise}}$ while

keeping the discount factor fixed at $\gamma = 0.99$:

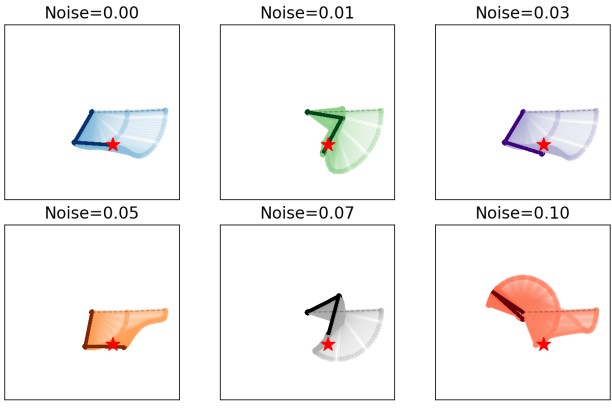

*Figure 5.* Example trajectories for visualizing how random action noise affects the policy.

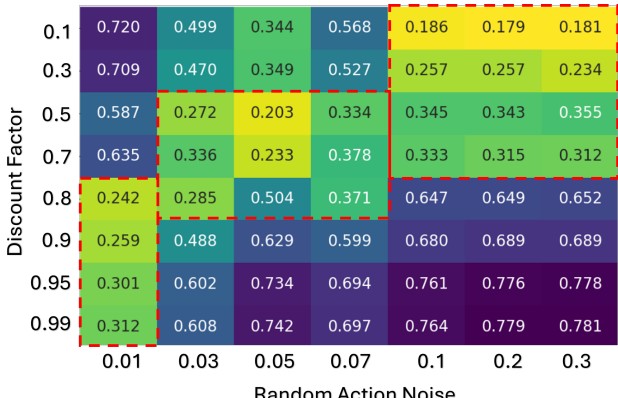

*Figure 6.* The JSD policy similarity between adding random noise with discount factor 0.99 and lower discount factors

Based on Figure 5, under minimal noise injection, the agent retains the same trajectory as the $\gamma = 0.99$. As the noise level increases, the trajectory loses stability. The arm exhibits a wide, sweeping motion around the center rather than a precise reach. The high variance in action prevents the agent from sufficiently damping the velocity, causing it to orbit the target rather than achieving a static equilibrium.

### 8.2.3. POLICY SIMILARITY

To quantify the similarity between noise-injected and discount-reduced policies, we cannot directly compare the actions taken by the policies since different policies can be unreliable at out-of-distribution states. We instead consider the distribution of reward received by both policies over time. Specifically, we measure the Jensen-Shannon Divergence (JSD) between the normalized reward distributions $P$ (discounted) and $Q$ (noisy) via their mixture $W = \frac{1}{2}(P + Q)$:

$$\text{JSD}(P \parallel Q) = \sqrt{0.5 D_{\text{KL}}(P \parallel W) + 0.5 D_{\text{KL}}(Q \parallel W)}.$$

Figure 6 confirms that noise-induced instability statistically mirrors temporal myopia across three regimes. Minimal noise ($\sigma \approx 0.01$) yields the low divergence characteristic of stable, high-$\gamma$ policies. Intermediate noise ($0.03 \leq \sigma \leq 0.07$) reproduces the oscillatory "wrangling" of mid-range discounts. Finally, high noise ($\sigma \geq 0.1$) produces reward distributions similar to the rotational failure of low-$\gamma$ policies. Additional experiments validating Theorem 5.1 via reset noise are listed in Appendix D, where we have provided a way to show the equivalence between the noise level and the discount factor.

## 9. Conclusion

The paper studied the role of environmental noise as a structural regularizer in MDPs. Our results suggest a reinterpretation of a common practice in reinforcement learning: using a smaller discount factor to cope with uncertainty in the environment that is not explicitly modeled. More broadly, the work highlights the discount factor as a proxy for structural uncertainty in the environment, rather than solely a preference over future rewards. By limiting the effective planning horizon, environmental stochasticity can allow for shallower reasoning, offering a route to interpretability that emerges naturally rather than being enforced by design.

## Impact Statement

Our work is closely related to interpretable reinforcement learning, which is important for safe and ethical AI.

## Acknowledgements

This material is based upon work supported by the National Science Foundation Graduate Research Fellowship Program under Grant No. DGE-2139754. This work was also supported by ARO grant #W911NF2210251 and the National Institutes of Health under R01DA054994. Any opinions, findings, and conclusions or recommendations expressed in this material are those of the author(s) and do not necessarily reflect the views of the NSF, ARO or NIH.

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

# A. Proofs

## A.1. Sticky and Reset Noise

**Theorem 4.1** (Sticky Noise). *For an MDP $M = (S, A, P, R)$ and its sticky counterpart $M_\beta$ with sticky parameter $\beta \in [0, 1]$, let $\pi^*$ be the optimal policy for $M_\beta$ under a discount factor $\gamma \in (0, 1)$. Then $\pi^*$ is also the optimal policy for the original MDP $M$ under a modified discount factor $\gamma' = \frac{\beta\gamma}{1-\gamma(1-\beta)} < \gamma$.*

*Proof.* The fixed point for the value function for policy $\pi$ in $M_\beta$ satisfies:

$$
\begin{aligned}
V^\pi_{\gamma,\beta} &= R + \gamma P^\pi_\beta V^\pi_{\gamma,\beta} \\
V^\pi_{\gamma,\beta} &= R + \gamma(\beta P^\pi + (1-\beta)I)V^\pi_{\gamma,\beta} \\
V^\pi_{\gamma,\beta} - \gamma(1-\beta)V^\pi_{\gamma,\beta} - \beta\gamma P^\pi V^\pi_{\gamma,\beta} &= R \\
(1 - \gamma(1-\beta))V^\pi_{\gamma,\beta} - \beta\gamma P^\pi V^\pi_{\gamma,\beta} &= R \\
V^\pi_{\gamma,\beta} - \frac{\beta\gamma}{(1-\gamma(1-\beta))}P^\pi V^\pi_{\gamma,\beta} &= \frac{R}{(1-\gamma(1-\beta))} \\
\left(I - \frac{\beta\gamma}{(1-\gamma(1-\beta))}P^\pi\right)V^\pi_{\gamma,\beta} &= \frac{R}{(1-\gamma(1-\beta))} \\
V^\pi_{\gamma,\beta} &= \left(I - \frac{\beta\gamma}{(1-\gamma(1-\beta))}P^\pi\right)^{-1}\frac{R}{(1-\gamma(1-\beta))}.
\end{aligned}
$$

Observe that this is equivalent to solving for the value function of $M$ *with the original transition matrix* but replacing $\gamma$ with with a scaled discount factor, $\frac{\beta\gamma}{(1-\gamma(1-\beta))}$ and reward scaled by $\frac{1}{(1-\gamma(1-\beta))}$. This can be replaced with the unscaled reward function without changing the optimal policy, completing the proof. $\square$

**Theorem 4.2** (Generalized Sticky Noise). *Consider $P_{\text{noisy}}$ and $P_{-\text{noisy}}$ as defined above, $P_\beta$ the original transition model with sticky noise $\beta$ added, and $\gamma' = \frac{\beta\gamma}{1-\gamma(1-\beta)}$. Define the following policies: $\pi^*_\gamma$ is optimal for the original MDP, $\pi^*_{\gamma'}$ is optimal for the original MDP with discount $\gamma'$, $\pi^*_{\gamma,\text{noisy}}$ is optimal for $P_{\text{noisy}}$ and $\gamma$, $\pi^*_{\gamma',-\text{noisy}}$ is optimal for $P_{-\text{noisy}}$ and $\gamma'$. Then we have:*

$$
V^{\pi^*_{\gamma',-\text{noisy}}}_{\gamma,\text{noisy}} = V^{\pi^*_{\gamma,\text{noisy}}}_{\gamma,\text{noisy}} \leq V^{\pi^*_{\gamma'}}_{\gamma,\beta} \leq V^{\pi^*_\gamma}_\gamma.
$$

We justify the three relationships in the theorem working from right to left:

- Due to the self transition assumption, every policy in the $\beta$-sticky version of the original MDP corresponds to a stochastic policy in the original MDP. Therefore, the optimal policy in the $\beta$-sticky MDP (which is the same as the optimal policy in the $\gamma'$-discounted version of the original MDP), cannot have higher value than the optimal policy in the original MDP. Thus, $V^{\pi^*_{\gamma'}}_{\gamma,\beta} \leq V^{\pi^*_\gamma}_\gamma$.

- Due to the convex noise assumption, the controllable self transition assumption, and the assumption that every action in the noisy MDP has a stickiness of at least $\beta$, every stochastic policy in the noisy MDP is a stochastic policy in the $\beta$-sticky MDP. Therefore, the value of the optimal policy in the noisy MDP cannot be larger than the value of the optimal policy in the sticky MDP. Thus, $V^{\pi^*_{\gamma,\text{noisy}}}_{\gamma,\text{noisy}} \leq V^{\pi^*_{\gamma'}}_{\gamma,\beta}$.

- Finally, observe that $P_{\text{noisy}}$ is a sticky version of $P_{-\text{noisy}}$ with stickiness $\beta$. This means that the optimal policy for $P_{-\text{noisy}}$ with discount $\gamma'$ must be the same as the optimal policy for $P_{\text{noisy}}$ with discount $\gamma$. Thus, $V^{\pi^*_{\gamma',-\text{noisy}}}_{\gamma,\text{noisy}} = V^{\pi^*_{\gamma,\text{noisy}}}_{\gamma,\text{noisy}}$.

The self transition and convex noise assumptions were used only for the upper bounds. If these assumptions are dropped but the minimum level of stickiness is preserved, then the equality justified in the final bullet above still holds.

**Theorem 5.1** (Reset Noise). *For an MDP $M = (S, A, P, R)$ and its reset counterpart $M_\alpha$, let $\pi^*$ be the optimal policy for $M_\alpha$ under a discount factor $\gamma \in (0, 1)$. Then $\pi^*$ is also the optimal policy for the original MDP $M$ under discount factor $\gamma' = \alpha\gamma < \gamma$.*

*Proof.* The value function for a policy in this $M_\alpha$ with discount factor $\gamma$ satisfies:

$$
\begin{aligned}
V^\pi_{\gamma,\alpha} &= R + \gamma P^\pi_\alpha V^\pi_{\gamma,\alpha} \\
V^\pi_{\gamma,\alpha} &= R + \gamma(\alpha P^\pi + (1-\alpha)P^\mu)V^\pi_{\gamma,\alpha} \\
V^\pi_{\gamma,\alpha} &= R + \gamma\alpha P^\pi V^\pi_{\gamma,\alpha} + \gamma(1-\alpha)P^\mu V^\pi_{\gamma,\alpha} \\
V^\pi_{\gamma,\alpha} - \gamma\alpha P^\pi V^\pi_{\gamma,\alpha} - \gamma(1-\alpha)P^\mu V^\pi_{\gamma,\alpha} &= R \\
(I - \gamma\alpha P^\pi - \gamma(1-\alpha)P^\mu)V^\pi_{\gamma,\alpha} &= R \\
V^\pi_{\gamma,\alpha} &= (I - \gamma\alpha P^\pi - \gamma(1-\alpha)P^\mu)^{-1}R
\end{aligned}
$$

Before simplifying this further, we review the Sherman-Morrison formula:

$$
(D + uv^T)^{-1} = D^{-1} - \frac{D^{-1}uv^T D^{-1}}{1 + v^T D^{-1}u}
$$

To align with our matrix inversion problem, we consider:

- $D = (I - \gamma\alpha P^\pi)$

- $u = -\gamma(1-\alpha)\mathbf{1}_n$

- $v = \mu$

Given this, we observe that:

- $D\mathbf{1}_n = (1 - \gamma\alpha)\mathbf{1}_n \implies D^{-1}\mathbf{1}_n = \frac{1}{1-\gamma\alpha}\mathbf{1}_n$

- $uv^T = -\gamma(1-\alpha)P^\mu$

- $D^{-1}u = -\gamma(1-\alpha)D^{-1}\mathbf{1}_n = -\gamma\frac{1-\alpha}{1-\gamma\alpha}\mathbf{1}_n$

- $v^T D^{-1}u = -\gamma\frac{1-\alpha}{1-\gamma\alpha}\mu^T\mathbf{1}_n = -\gamma\frac{1-\alpha}{1-\gamma\alpha}$

- $1 + v^T D^{-1}u = 1 - \gamma\frac{1-\alpha}{1-\gamma\alpha} = \frac{1-\gamma\alpha-\gamma+\gamma\alpha}{1-\gamma\alpha} = \frac{1-\gamma}{1-\gamma\alpha}$

- $D^{-1}uv^T = -\gamma\frac{1-\alpha}{1-\gamma\alpha}\mathbf{1}_n\mu^T = -\gamma\frac{1-\alpha}{1-\gamma\alpha}P^\mu$

- $\frac{D^{-1}uv^T}{1+v^T D^{-1}u} = \frac{-\gamma\frac{1-\alpha}{1-\gamma\alpha}P^\mu}{\frac{1-\gamma}{1-\gamma\alpha}} = -\gamma\frac{1-\alpha}{(1-\gamma)}P^\mu.$

Using the Sherman-Morrison formula, we get:

$$
\begin{aligned}
V^\pi_{\gamma,\alpha} &= (I - \gamma\alpha P^\pi - \gamma(1-\alpha)P^\mu)^{-1}R \\
&= (D + uv^T)^{-1}R \\
&= \left(D^{-1} - \frac{D^{-1}uv^T D^{-1}}{1 + v^T D^{-1}u}\right)R \\
&= D^{-1}R - \frac{D^{-1}uv^T}{1 + v^T D^{-1}u}D^{-1}R \\
&= (I - \gamma\alpha P^\pi)^{-1}R + \gamma\frac{1-\alpha}{1-\gamma}P^\mu(I - \gamma\alpha P^\pi)^{-1}R.
\end{aligned}
$$

Observe that $(I - \gamma\alpha P^\pi)^{-1}R$ is the value function of $\pi$ in the original MDP with discount factor $\gamma\alpha$:

$$
V^\pi_{\gamma\alpha} = (I - \gamma\alpha P^\pi)^{-1}R.
$$

Note that the subscript to the value function above is not an ordered pair, indicating that it is the value in the *original* MDP. This allows us to rewrite the value function for the noisy MDP as:

$$
\begin{aligned}
V_{\gamma,\alpha}^{\pi} &= V_{\gamma\alpha}^{\pi} + \gamma \frac{1-\alpha}{1-\gamma} P^{\mu} V_{\gamma\alpha}^{\pi} \\
&= V_{\gamma\alpha}^{\pi} + \gamma \frac{1-\alpha}{1-\gamma} \overline{v_{\gamma\alpha}^{\pi}}
\end{aligned}
$$

where $\overline{v_{\gamma\alpha}^{\pi}}$ is the $\mu$-weighted mean state value in $V_{\gamma\alpha}^{\pi}$.

So far, we have made a connection between policy values in the noisy MDP with discount $\gamma$ and policy value in the original MDP with reduced discount $\gamma\alpha$, but it is not (yet) clear that they might share the same optimal policy. Since $\overline{v_{\gamma\alpha}^{\pi}}$ is a policy-dependent offset, we still need to show that the offset does not change which policy is optimal. To see this, we first make the following observation about about any average of state values for two policies. Let $\rho$ be a column vector corresponding to a distribution over states. Then for any $\pi_1$ and $\pi_2$, where $\pi_1$ weakly dominates $\pi_2$, and positive constant $c$, we have:

$$
V^{\pi_1} \geq V^{\pi_2} \rightarrow c\rho^T V^{\pi_1} \geq c\rho^T V^{\pi_2}.
$$

Now, consider $\rho = \mu$, and $c = \gamma\frac{1-\alpha}{1-\gamma}$. Assume $\pi^*$ is optimal for the original MDP with discount factor $\gamma\alpha$ and let $\pi$ be any other policy:

$$
\begin{aligned}
V_{\gamma\alpha}^{\pi^*} &\geq V_{\gamma\alpha}^{\pi} \\
V_{\gamma\alpha}^{\pi^*} + c\mu^T V_{\gamma\alpha}^{\pi^*} &\geq V_{\gamma\alpha}^{\pi} + c\mu^T V_{\gamma\alpha}^{\pi^*} \\
V_{\gamma\alpha}^{\pi^*} + c\overline{v_{\gamma\alpha}^{\pi^*}} &\geq V_{\gamma\alpha}^{\pi} + c\mu^T V_{\gamma\alpha}^{\pi} \\
V_{\gamma\alpha}^{\pi^*} + c\overline{v_{\gamma\alpha}^{\pi^*}} &\geq V_{\gamma\alpha}^{\pi} + c\overline{v_{\gamma\alpha}^{\pi}} \\
V_{\gamma,\alpha}^{\pi^*} &\geq V_{\gamma,\alpha}^{\pi}.
\end{aligned}
$$

Therefore, $\pi^*$ must also be optimal for $M_\alpha$ with discount $\gamma$. $\qquad\square$

## A.2. Slowly-Changing Reversible Noise

To prove Lemma 6.2, we provide a slightly different, though less intuitive, bound that we will later show is, in fact, stronger through Lemma A.3. Let $c(x) = \frac{2}{1+\sqrt{1-4x}}$, where we note that $c$ is the generating function of the Catalan numbers.

**Lemma A.1** (Stronger Lemma 6.2). *For MDP $M = (S, A, P, R)$ and a $(C_{\min}, C_{\max})$-slowly-changing reversible noisy MDP $M_\epsilon = (S, A, P^{\text{noisy}}, R)$, define $\gamma' < \gamma$ to be $\gamma' := \gamma(1-\epsilon)c(\gamma^2\epsilon(1-\epsilon)) = \frac{2\gamma(1-\epsilon)}{1+\sqrt{1-4\gamma^2\epsilon(1-\epsilon)}}$. For every policy $\pi$ on $M$, there exists a modified nonstationary policy $\pi'$ satisfying*

$$
-\frac{C_{\min}}{1-\gamma}\left(\frac{\gamma'}{1-\gamma'} - (1-2\epsilon)\cdot\frac{\gamma}{1-\gamma}\right) \leq V_{\gamma,\text{noisy}}^{\pi'} - \frac{1-\gamma'}{1-\gamma}V_{\gamma'}^{\pi} \leq \frac{C_{\max}}{1-\gamma}\left(\frac{\gamma'}{1-\gamma'} - (1-2\epsilon)\cdot\frac{\gamma}{1-\gamma}\right). \quad (3)
$$

*Additionally, for any values of $C_{\min}$, $C_{\max}$, $\gamma$, and $\epsilon$, there exists an MDP $M = (S, A, P, R)$ and a $(C_{\min}, C_{\max})$-slowly-changing reversible noisy MDP $M_\epsilon = (S, A, P^{\text{noisy}}, R)$ such that either side of Inequality 3 holds with equality.*

*Proof.* For a given policy $\pi$, consider the altered nonstationary policy $\pi'$ that attempts to reverse the effects of noise. We make the assumption that $\pi'$ can depend on previous states visited as well as whether a noise activation occurred or not, neither of which allows $\pi'$ to perform better than an optimal stationary policy. $\pi'$ follows the actions that $\pi$ would take until noise takes effect. Then, $\pi'$ will attempt to return to the original state from which the noisy outcome occurred. If noise takes effect consecutively, then $\pi'$ will continue to reverse the effects of noise until successful, which is possible due to reversibility. We note that $\pi'$ does not attempt to exploit noise in any way, such as by changing its behavior if noise takes it to a better state than where $\pi$ would aim for. This is why assuming that knowledge of when noise has occurred is not problematic. In practice, an agent would take the action with the highest expected value in the discounted MDP and would do at least as well as $\pi'$.

We use the return of policy $\pi$ in the original MDP to bound the return of $\pi'$ in the noisy MDP. Suppose that in the original MDP, the policy $\pi$ visits state $s_j$ and collects reward $r_j$ at time $j$, so that $V_\gamma^\pi(s_0) = \mathbb{E}[\sum_{j=0}^\infty \gamma^j r_j] = \sum_{j=0}^\infty \gamma^j \mathbb{E}[r_j]$. Now let's focus on the noisy MDP with policy $\pi'$. We first couple the randomness of the trajectories of $\pi$ and $\pi'$ by supposing that, when $\pi'$ is transitioning from state $s_j$ to $s_{j+1}$ when following $\pi$, the randomness of the non-noisy component of the transition is the same for both policies. That is, if noise does not activate, then policy $\pi'$ will reach the same state $s_{j+1}$. Denote by $i_j$ the time the agent arrived at state $s_j$ after visiting all previous states and reversing noise, and denote by $\Delta_j$ the number of time steps the agent was stranded trying to go from state $s_j$ to state $s_{j+1}$. Thus, $i_{j+1} = i_j + \Delta_j$. Note that each $\Delta_j$ is i.i.d. and solely depends on the activations of noise.

Suppose that, starting at time $t = 0$, $\pi'$ attempts to transition from state $s_j$ to state $s_{j+1}$. At time $t$, we can define $X_t^j$ to be the number of times that noise has activated minus the number of times that noise has not activated. We can see that $\{X_t^j\}_{t=0}^\infty$ forms a biased random walk where $X_0^j = 1$, $X_{t+1}^j = X_t^j + 1$ with probability $\epsilon$, $X_{t+1}^j = X_t^j - 1$ with probability $1 - \epsilon$, and the process ends once $X_t^j = 0$ since $\pi'$ will have successfully reached state $s_{j+1}$. In particular, $\{X_t^j\}_{t=0}^{\Delta_j}$ forms a biased random walk with parameter $\epsilon$, and $\Delta_j$ is the time it takes for the random walk to return to $0$[4]. Note that $P(\Delta_j < \infty) = 1$ since $\epsilon \leq 0.5$. We can see that the reward that $\pi'$ collects at time $t < \Delta_j$ is at most $r_j + (X_t^j - 1)C_{\max}$ since the MDP is slowly-changing and the state at time $t$ is at most $X_t^j - 1$ steps away from the state $s_j$.

From the argument above, the value function $V_{\gamma,\text{noisy}}^{\pi'}$ of the noisy MDP is bounded above by the expectation by the following:

$$V_{\gamma,\text{noisy}}^{\pi'}(s_0) \leq \sum_{j=0}^\infty \mathbb{E}\left[\sum_{t=0}^{\Delta_j - 1} \gamma^{i_j + t}(r_j + (X_t^j - 1)C_{\max})\right] \tag{4}$$

Noting that $r_j$, $i_j$, and $(\Delta_j, X_t^j)$ are independent, we can now write the above quantity as $r + C$, where $r$ and $C$ are defined as follows:

$$r = \sum_{j=0}^\infty \mathbb{E}[\gamma^{i_j}]\mathbb{E}[r_j]\mathbb{E}\left[\sum_{t=0}^{\Delta_j - 1} \gamma^t\right]$$

$$C = C_{\max} \sum_{j=0}^\infty \mathbb{E}[\gamma^{i_j}]E\left[\sum_{t=0}^{\Delta_j - 1} \gamma^t(X_t^j - 1)\right].$$

We first examine $r$. Since each $\Delta_j$ is independent, we have that $\mathbb{E}[\gamma^{i_j}] = \mathbb{E}[\gamma^{\Delta_0} \cdot \ldots \cdot \gamma^{\Delta_{j-1}}] = \mathbb{E}[\gamma^{\Delta_0}] \cdot \ldots \cdot \mathbb{E}[\gamma^{\Delta_{j-1}}]$. Furthermore, since each $\Delta_j$ is identically distributed, let $\Delta$ come from the same common distribution. Then, we have that

$$r = \sum_{j=0}^\infty \mathbb{E}[\gamma^{i_j}]\mathbb{E}[r_j]\mathbb{E}\left[\sum_{t=0}^{\Delta_j - 1} \gamma^t\right]$$

$$= \sum_{j=0}^\infty (\mathbb{E}[\gamma^{\Delta_0}] \cdot \ldots \cdot \mathbb{E}[\gamma^{\Delta_{j-1}}])\mathbb{E}[r_j]\mathbb{E}\left[\frac{1 - \gamma^{\Delta_j}}{1 - \gamma}\right]$$

$$= \frac{1 - \mathbb{E}[\gamma^\Delta]}{1 - \gamma} \sum_{j=0}^\infty \mathbb{E}[\gamma^\Delta]^j \mathbb{E}[r_j].$$

We can see that $r$ is exactly equal to $\frac{1-\mathbb{E}[\gamma^\Delta]}{1-\gamma} V_{\mathbb{E}[\gamma^\Delta]}^\pi(s_0)$. As we will next prove, $\mathbb{E}[\gamma^\Delta]$ is precisely $\gamma'$, which shows that $r = \frac{1-\gamma'}{1-\gamma} V_{\gamma'}^\pi(s_0)$.

To compute $\mathbb{E}[\gamma^\Delta]$, we examine the PMF of $\Delta$. Clearly, $\Delta$ is odd, so consider $P(\Delta = 2k + 1)$. It is well-known that the number of paths in the random walk that return to $0$ at time $2k + 1$ is precisely the $k$th Catalan number $C_k$. Each such path goes up $k$ times and down $k + 1$ times, so each path is taken with probability $\epsilon^k(1 - \epsilon)^{k+1}$. Thus,

---

[4]This is precisely a Gambler's Ruin problem, from which we borrow some machinery.

$P(\Delta = 2k+1) = C_k \epsilon^k (1-\epsilon)^{k+1}$. This implies that

$$\mathbb{E}[\gamma^\Delta] = \gamma \sum_{k=0}^{\infty} P(\Delta = 2k+1)\gamma^{2k} = \gamma(1-\epsilon) \sum_{k=0}^{\infty} C_k \left(\gamma^2 \epsilon(1-\epsilon)\right)^k.$$

We can see that the summation is precisely the generating function of the Catalan numbers evaluated at $\gamma^2 \epsilon(1-\epsilon)$. Since the generating function is known to be $c(x) = \frac{2}{1+\sqrt{1-4x}}$, we have a closed form

$$\mathbb{E}[\gamma^\Delta] = \gamma(1-\epsilon)c(\gamma^2 \epsilon(1-\epsilon)) = \gamma'.$$

Lastly, we compute $C$. To do this, we examine a case where Inequality (4) holds with equality for all states. Since $C$ does not depend on the underlying MDP, it suffices to compute $C$ for this case, which we can do by computing $V_{\gamma,\text{noisy}}^{\pi'}$ and $V_{\gamma'}^{\pi}$ and writing $C = V_{\gamma,\text{noisy}}^{\pi'}(s) - \frac{1-\gamma'}{1-\gamma}V_{\gamma'}^{\pi}(s)$ for any state $s \in S$. As a corollary, our bound will be tight for this MDP.

Consider the MDP with states $S = \{S_i\}_{i \in \mathbb{Z}}$, where state $S_i$ has reward $C_{\max} \cdot i$. Furthermore, each state has exactly one action that deterministically moves the agent from state $S_i$ to state $S_{i-1}$. When noise activates, the agent is instead deterministically moved from state $i$ to state $i+1$. The resulting noisy MDP is $(0, C_{\max})$-slowly-changing and reversible. There is only a single policy $\pi$, so $\pi$ and $\pi'$ are equivalent for this MDP. Clearly, Inequality (4) holds with equality since, when attempting to transition from the $j$th state to the $(j+1)$th state with respect to $\pi$ (not to be confused with states $S_j$ and $S_{j+1}$), the reward collected at time $t$ while stranded is exactly $r_j + (X_t^j - 1) \cdot C_{\max}$.

Suppose that the agent begins at state $S_x$. We can see that the expected position of the agent at time $i$ in the noisy MDP is $x + i(\epsilon - (1-\epsilon)) = x + i(2\epsilon - 1)$. Since the position corresponds linearly to the reward, this implies that the expected reward collected at time $i$ is $C_{\max}(x + i(2\epsilon - 1))$. Thus, the value of state $S_x$ in the noisy MDP (for the only policy) is

$$V_{\gamma,\text{noisy}}^{\pi'}(S_x) = C_{\max} \sum_{i=0}^{\infty} \gamma^i(x + i(2\epsilon - 1)) = C_{\max} \cdot \frac{x}{1-\gamma} + C_{\max}(2\epsilon - 1) \cdot \frac{\gamma}{(1-\gamma)^2}.$$

Setting $\epsilon = 0$, we have that, without noise and with discount factor $\gamma'$, the value of state $x$ is

$$V_{\gamma'}^{\pi}(S_x) = C_{\max} \cdot \frac{x}{1-\gamma'} - C_{\max} \cdot \frac{\gamma'}{(1-\gamma')^2}.$$

This means that

$$C = V_{\gamma,\text{noisy}}^{\pi'}(S_x) - \frac{1-\gamma'}{1-\gamma}V_{\gamma'}^{\pi}(S_x) = \frac{C_{\max}}{1-\gamma}\left(\frac{\gamma'}{1-\gamma'} - (1-2\epsilon) \cdot \frac{\gamma}{1-\gamma}\right)$$

Thus, we have proven that, for all MDPs, $V_{\gamma,\text{noisy}}^{\pi'} - \frac{1-\gamma'}{1-\gamma}V_{\gamma'}^{\pi} \leq \frac{C_{\max}}{1-\gamma}\left(\frac{\gamma'}{1-\gamma'} - (1-2\epsilon) \cdot \frac{\gamma}{1-\gamma}\right)$. By negating every reward, we can show the other inequality as well.

$\square$

We next prove Lemma 6.3. We again provide a more general result by relaxing the bound and allow $\gamma'$ to be any discount factor smaller than $\gamma$.

**Lemma A.2** (Stronger Lemma 6.3). *For MDP $M = (S, A, P, R)$, consider any pair of discount factors $\gamma$ and $\gamma'$ with $\gamma' < \gamma$. Let $t^*$ be the largest time $t$ for which $\left(\frac{\gamma}{\gamma'}\right)^t < \frac{1-\gamma'}{1-\gamma}$. For any policy $\pi$, it holds that*

$$V_\gamma^\pi(s) \leq \frac{1-\gamma'}{1-\gamma}V_{\gamma'}^\pi(s) + \frac{(\gamma - \gamma')\gamma^{t^*}}{(1-\gamma)(1-\gamma')}$$

*for all $s \in S$.*

*Proof.* Let $r_t$ denote the stochastic reward collected at time $t$ by policy $\pi$. We can write the difference between the two scaled values as

$$V_\gamma^\pi(s) - \frac{1-\gamma'}{1-\gamma}V_{\gamma'}^\pi(s) = \sum_{t=0}^{\infty}\left(\gamma^t - \frac{1-\gamma'}{1-\gamma} \cdot (\gamma')^t\right)E[r_t] = \sum_{t=0}^{\infty}(\gamma')^t\left(\left(\frac{\gamma}{\gamma'}\right)^t - \frac{1-\gamma'}{1-\gamma}\right)E[r_t].$$

From this formula, we can see that the worst-case is when $E[r_t] = 0$ when $(\frac{\gamma}{\gamma'})^t \leq \frac{1-\gamma'}{1-\gamma}$ and $E[r_t] = 1$ otherwise. Since $(\frac{\gamma}{\gamma'})^t$ is monotonically increasing, these two cases occur precisely when $t \leq t^*$ and $t > t^*$ respectively, where $t^* = \lceil \frac{\log(1-\gamma')-\log(1-\gamma)}{\log \gamma - \log \gamma'} \rceil$. We thus have that

$$V_\gamma^\pi(s) - \frac{1-\gamma'}{1-\gamma}V_{\gamma'}^\pi(s) \leq \sum_{t=t^*+1}^\infty \left( \gamma^t - \frac{1-\gamma'}{1-\gamma} \cdot (\gamma')^t \right)$$
$$= \frac{\gamma^{t^*+1} - (\gamma')^{t^*+1}}{1-\gamma}$$
$$\leq \frac{\gamma^{t^*+1} - \frac{1-\gamma}{1-\gamma'} \cdot \gamma^{t^*} \cdot \gamma'}{1-\gamma}$$
$$= \gamma^{t^*} \cdot \frac{\gamma - \gamma'}{(1-\gamma)(1-\gamma')}.$$

$\square$

We next present the proof for Lemma A.3.

**Lemma A.3.** *For a fixed $\epsilon \in [0, 0.5)$ and $\gamma \in [0, 1)$, let $\gamma' = \gamma(1-\epsilon)c(\gamma^2\epsilon(1-\epsilon))$. Then, $\frac{1}{1-\gamma}\left( \frac{\gamma'}{1-\gamma'} - (1-2\epsilon) \cdot \frac{\gamma}{1-\gamma} \right) \leq \frac{\epsilon\gamma}{1-2\epsilon} \cdot \frac{1}{1-\gamma}$, and $\frac{(\gamma-\gamma')\gamma^{t^*}}{(1-\gamma)(1-\gamma')} \leq 2\epsilon(1-2\epsilon)^{\frac{1-2\epsilon}{2\epsilon}} \cdot \frac{1}{1-\gamma}$ where $t^*$ is the largest integer $t$ for which $\left(\frac{\gamma}{\gamma'}\right)^t < \frac{1-\gamma'}{1-\gamma}$.*

*Proof.* We break this proof into steps since there are multiple properties of various functions of $\gamma$, $\gamma'$, and $\epsilon$ that we must show and use as lemmas for following properties. We first show that $\frac{1-\gamma'}{1-\gamma} \to \frac{1}{1-2\epsilon}$. We have that $\gamma' \to 1$ since

$$c(\gamma^2\epsilon(1-\epsilon)) = \frac{1-\sqrt{1-4\gamma^2\epsilon(1-\epsilon)}}{2\gamma^2\epsilon(1-\epsilon)} \to \frac{1-\sqrt{1-4\epsilon(1-\epsilon)}}{2\epsilon(1-\epsilon)} = \frac{1-\sqrt{(1-2\epsilon)^2}}{2\epsilon(1-\epsilon)} = \frac{1}{1-\epsilon},$$

so $\gamma' = \gamma(1-\epsilon)c(\gamma^2\epsilon(1-\epsilon)) \to 1$. Next, by L'Hopital's rule,

$$\lim_{\gamma \to 1} \frac{1-\gamma'}{1-\gamma} = \lim_{\gamma \to 1}[(1-\epsilon)c(\gamma^2\epsilon(1-\epsilon)) + 2\gamma^2\epsilon(1-\epsilon)^2c'(\gamma^2\epsilon(1-\epsilon))] = 1 + 2\epsilon(1-\epsilon)^2c'(\epsilon(1-\epsilon)).$$

We can compute that $c'(x) = \frac{4}{(1+\sqrt{1-4x})^2\sqrt{1-4x}}$, so $c'(\epsilon(1-\epsilon)) = \frac{1}{(1-\epsilon)^2(1-2\epsilon)}$. Thus, $\lim_{\gamma \to 1}\frac{1-\gamma'}{1-\gamma} = \frac{1}{1-2\epsilon}$.

Next, we show that the same map $\gamma \mapsto \frac{1-\gamma'}{1-\gamma}$ is increasing. We do this by exploiting the power series representation of this function. Since $c(x)$ is the generating function of the Catalan numbers $C_k$, we can write

$$\frac{1-\gamma'}{1-\gamma} = \frac{1-\gamma(1-\epsilon)c(\gamma^2\epsilon(1-\epsilon))}{1-\gamma} = \left(\sum_{k=0}^\infty \gamma^k\right)\left(1 - (1-\epsilon)\sum_{k=0}^\infty C_k\epsilon^k(1-\epsilon)^k\gamma^{2k+1}\right).$$

After expanding and grouping like terms together, we have that $\frac{1-\gamma'}{1-\gamma} = \sum_{k=0}^\infty a_k\gamma^k$ where $a_k = 1 - (1-\epsilon)\sum_{m=0}^{\lfloor \frac{k-1}{2} \rfloor} C_m\epsilon^m(1-\epsilon)^m$. We can notice that $a_k \geq 1 - (1-\epsilon)\sum_{m=0}^\infty C_m\epsilon^m(1-\epsilon)^m = 1 - (1-\epsilon)c(\epsilon(1-\epsilon)) = 0$. Thus, we have that $\frac{d}{d\gamma}\frac{1-\gamma'}{1-\gamma} = \sum_{k=1}^\infty ka_k\gamma^{k-1} \geq 0$, so the function is increasing.

We now show that the function $\frac{\log \gamma'}{\log \gamma}$ is increasing. For ease of notation, let $A(\gamma) = \gamma' = \gamma(1-\epsilon)c(\gamma^2\epsilon(1-\epsilon))$. If $\gamma = e^t$ for $t \in (-\infty, 0)$, then it suffices to show that the function $t \mapsto \frac{\log A(e^t)}{t}$ is increasing. Note that $\log A(e^0) = 0$. Then, it is enough to show that $\log A(e^t)$ is convex. This is because, for any $t_1 < t_2 < 0$, we can define $\lambda = \frac{t_2}{t_1}$ so that $t_2 = \lambda t_1 + (1-\lambda) \cdot 0$. By convexity, we have that

$$\log A(e^{t_2}) \leq \lambda A(e^{t_1}) + (1-\lambda)A(e^0) = \frac{t_2}{t_1}A(e^{t_1}).$$

Rearranging terms produces that $\frac{\log A(e^t)}{t}$ is increasing as desired. To show that $\log A(e^t)$ is convex, we simply differentiate twice to see that

$$\frac{d^2}{dt^2} \log A(e^t) = \frac{4\epsilon A(e^t)e^t}{1-\epsilon} \cdot \frac{A'(e^t)(2 - c(e^{2t}\epsilon(1-\epsilon))) + \epsilon(1-\epsilon)A(e^t) \cdot c'(e^{2t}\epsilon(1-\epsilon))e^t}{(2 - c(e^{2t}\epsilon(1-\epsilon)))^2}.$$

Since $A(\gamma)$ and $c(x)$ are both increasing functions and $c(x) \in [1, 2]$, we can see that the second derivative is nonnegative and therefore $\log A(e^t)$ is convex. Thus, $\frac{\log \gamma'}{\log \gamma}$ is increasing.

We have now proven enough properties to tackle the main lemma. To show that $b \leq \frac{\gamma}{1-\gamma} \cdot \frac{\epsilon}{1-2\epsilon}$, it is sufficient to notice that the map $\gamma \mapsto \frac{\gamma'}{1-\gamma'} - (1-2\epsilon) \cdot \frac{\gamma}{1-\gamma}$ is convex and tends to $\frac{\epsilon}{1-2\epsilon}$ as $\gamma \to 1$. The latter property can be shown by combining denominators to a single quotient and then applying L'Hopital's rule twice by taking the second derivative, similar to our earlier computation. The former property requires a little bit more work. Through a tedious algebraic expansion, the following equality can be shown:

$$\frac{\gamma'}{1-\gamma'} - (1-2\epsilon) \cdot \frac{\gamma}{1-\gamma} = \epsilon\gamma + \epsilon\gamma^2 \cdot \frac{1 - (1-\epsilon)c(\gamma^2\epsilon(1-\epsilon))}{1-\gamma}.$$

If $r(\gamma) = \frac{1 - (1-\epsilon)c(\gamma^2\epsilon(1-\epsilon))}{1-\gamma}$, then we note that $\frac{1-\gamma'}{1-\gamma} = 1 + \gamma r(\gamma)$. However, we showed earlier that every coefficient of the power series of $\frac{1-\gamma'}{1-\gamma}$ is nonnegative, so this implies that every coefficient of the power series of $r(\gamma)$ is also nonnegative. This then implies that every coefficient of the power series of $\frac{\gamma'}{1-\gamma'} - (1-2\epsilon) \cdot \frac{\gamma}{1-\gamma}$ is nonnegative. In particular, this function is convex in $\gamma$.

Lastly, we show that $\frac{(\gamma-\gamma')\gamma^{t^*}}{1-\gamma'} \leq 2\epsilon_1(1-2\epsilon_1)^{\frac{1-2\epsilon_1}{2\epsilon_1}}$ by first showing that $\frac{(\gamma-\gamma')\gamma^{t^*}}{1-\gamma'} \to 2\epsilon_1(1-2\epsilon_1)^{\frac{1-2\epsilon_1}{2\epsilon_1}}$ as $\gamma \to 1$ and then arguing that this function is increasing in $\gamma$. We first note that $\frac{\gamma-\gamma'}{1-\gamma'} = 1 - \frac{1-\gamma}{1-\gamma'} \to 2\epsilon$. We therefore focus on $t^* \log \gamma$ where $t^* = \lceil \frac{\log(1-\gamma') - \log(1-\gamma)}{\log\gamma - \log\gamma'} \rceil$. Using the fact that $\log(1-x) = -x + O(x^2)$, we have that

$$\lim_{\gamma\to1} \frac{(\log(1-\gamma') - \log(1-\gamma))\log\gamma}{\log\gamma - \log\gamma'} = \lim_{\gamma\to1} \frac{\log\frac{1-\gamma'}{1-\gamma}\log\gamma}{\log\frac{\gamma}{\gamma'}}$$

$$= -\log(1-2\epsilon)\lim_{\gamma\to1}\frac{\log\gamma}{\log\frac{\gamma}{\gamma'}}$$

$$= -\log(1-2\epsilon)\lim_{\gamma\to1}\frac{\gamma-1}{\frac{\gamma}{\gamma'}-1}$$

$$= -\log(1-2\epsilon)\lim_{\gamma\to1}\gamma' \cdot \frac{1}{1 - \frac{1-\gamma'}{1-\gamma}}$$

$$= -\log(1-2\epsilon) \cdot \frac{1}{1 - \frac{1}{1-2\epsilon}}$$

$$= \log(1-2\epsilon) \cdot \frac{1-2\epsilon}{2\epsilon}.$$

Thus, $\lim_{\gamma\to1}\frac{(\gamma-\gamma')\gamma^{t^*}}{1-\gamma'} = 2\epsilon \exp(\log(1-2\epsilon) \cdot \frac{1-2\epsilon}{2\epsilon}) = 2\epsilon(1-2\epsilon)^{\frac{1-2\epsilon}{2\epsilon}}$. Next, to show that $\frac{(\gamma-\gamma')\gamma^{t^*}}{1-\gamma'}$ is increasing, we take the derivative of its logarithm. This gives

$$\frac{d}{d\gamma}\log\frac{(\gamma-\gamma')\gamma^t}{1-\gamma'} = \frac{d}{d\gamma}\left(\log(\gamma-\gamma') - \log(1-\gamma') + (\log(1-\gamma') - \log(1-\gamma)) \cdot \frac{\log\gamma}{\log\gamma - \log\gamma'}\right)$$

$$= \frac{1 - \frac{d}{d\gamma}\gamma'}{\gamma-\gamma'} + \frac{\frac{d}{d\gamma}\gamma'}{1-\gamma'} + \left(\frac{1}{1-\gamma} - \frac{\frac{d}{d\gamma}\gamma'}{1-\gamma'}\right) \cdot \frac{\log\gamma}{\log\gamma - \log\gamma'} + \log\frac{1-\gamma'}{1-\gamma} \cdot \frac{d}{d\gamma}\frac{\log\gamma}{\log\gamma - \log\gamma'}.$$

Since $\frac{1-\gamma'}{1-\gamma} > 1$ and $\frac{\log\gamma}{\log\gamma - \log\gamma'} = \frac{1}{1 - \frac{\log\gamma'}{\log\gamma}}$ is increasing, the last term is nonnegative. Furthermore, we know that $\frac{\log\gamma'}{\log\gamma} \geq \frac{1-\gamma'}{1-\gamma}$ since the function $x \mapsto \frac{\log x}{1-x}$ is increasing. We also know that $\frac{1}{1-\gamma} - \frac{\frac{d}{d\gamma}\gamma'}{1-\gamma'} = \frac{d}{d\gamma}\log\frac{1-\gamma'}{1-\gamma} \geq 0$ since $\frac{1-\gamma'}{1-\gamma}$ is

increasing. Combining these facts gives that

$$\frac{d}{d\gamma} \log \frac{(\gamma - \gamma')\gamma^t}{1 - \gamma'} \geq \frac{1 - \frac{d}{d\gamma}\gamma'}{\gamma - \gamma'} + \frac{\frac{d}{d\gamma}\gamma'}{1 - \gamma'} + \left( \frac{1}{1 - \gamma} - \frac{\frac{d}{d\gamma}\gamma'}{1 - \gamma'} \right) \cdot \frac{1}{1 - \frac{1 - \gamma'}{1 - \gamma}} = 0$$

and so $\frac{(\gamma - \gamma')\gamma^{t^*}}{1 - \gamma'}$ is increasing. Thus, $\frac{(\gamma - \gamma')\gamma^{t^*}}{1 - \gamma'} \leq 2\epsilon(1 - 2\epsilon)^{\frac{1 - 2\epsilon}{2\epsilon}}$. $\qquad \square$

From the above lemmas, we recover Theorem 6.1.

**Theorem A.4** (Stronger Theorem 6.1). *For MDP $M = (S, A, P, R)$ and a $(C_{\min}, C_{\max})$-slowly-changing reversible noisy MDP $M_\epsilon = (S, A, P^{\text{noisy}}, R)$, define $\gamma' < \gamma$ to be $\gamma' := \frac{2\gamma(1 - \epsilon)}{1 + \sqrt{1 - 4\gamma^2\epsilon(1 - \epsilon)}}$. Let $\pi^*_{\gamma'}$, $\pi^*_\gamma$, and $\pi^*_{\gamma,\text{noisy}}$ maximize $V^\pi_{\gamma'}$, $V^\pi_\gamma$, and $V^\pi_{\gamma,\text{noisy}}$ respectively. If $V^{\pi^*_{\gamma,\text{noisy}}}_{\gamma,\text{noisy}} \leq V^{\pi^*_\gamma}_\gamma$, then there exists a nonstationary policy $\pi'$ obtained from $\pi^*_{\gamma'}$ such that*

$$V^{\pi'}_{\gamma,\text{noisy}} \geq \frac{1 - \gamma'}{1 - \gamma} V^{\pi^*_{\gamma'}}_{\gamma'} - \frac{C_{\min}}{1 - \gamma} \left( \frac{\gamma'}{1 - \gamma'} - (1 - 2\epsilon) \cdot \frac{\gamma}{1 - \gamma} \right)$$

$$\geq V^{\pi^*_{\gamma,\text{noisy}}}_{\gamma,\text{noisy}} - \frac{C_{\min}}{1 - \gamma} \left( \frac{\gamma'}{1 - \gamma'} - (1 - 2\epsilon) \cdot \frac{\gamma}{1 - \gamma} \right) - \frac{(\gamma - \gamma')\gamma^{t^*}}{(1 - \gamma)(1 - \gamma')}$$

$$\geq V^{\pi^*_{\gamma,\text{noisy}}}_{\gamma,\text{noisy}} - \left( \frac{C_{\min}\epsilon\gamma}{1 - 2\epsilon} + 2\epsilon(1 - 2\epsilon)^{\frac{1 - 2\epsilon}{2\epsilon}} \right) \cdot \frac{1}{1 - \gamma}.$$

*Proof.* The first inequality follows from applying Lemma A.1 on $\pi^*$. The second inequality comes from applying Lemma A.2 on $\pi^*_\gamma$ and then using the inequality $V^{\pi^*_{\gamma'}}_{\gamma'} \geq V^{\pi^*_\gamma}_{\gamma'}$ and the assumption $V^{\pi^*_\gamma}_\gamma \geq V^{\pi^*_{\gamma,\text{noisy}}}_{\gamma,\text{noisy}}$. Lastly, the third inequality is due to a direct application of Lemma A.3. $\qquad \square$

### A.3. Tightrope MDP Example

**Theorem A.5.** *There exists an MDP $M$ with rewards strictly bounded in $[-1, 1]$, starting state $s_0$, and a noise model parameterized by $\epsilon$, such that:*

*Let $\Pi_{clean}$ be the set of all policies that are optimal in the noiseless version of $M$ for **any** discount factor $\gamma' \in (0, 1)$:*

$$\Pi_{clean} = \bigcup_{\gamma' \in (0,1)} \left\{ \arg\max_\pi V^{clean}_{\gamma'}(\pi) \right\}$$

*Then, for a specific noise parameter $\epsilon$:*

1. *For **all** discount factors $\gamma \in (0, 1)$, the optimal policy for the noisy MDP (denoted $\pi^*_{\text{noisy}}$) strictly dominates every policy in $\Pi_{clean}$:*

$$V^{\pi^*_{\text{noisy}}}_{\text{noisy}}(s_0) > V^\pi_{\text{noisy}}(s_0) \quad \forall \pi \in \Pi_{clean}$$

2. *Furthermore, for sufficiently large $\gamma$, this domination holds by a margin of at least $1/2$:*

$$V^{\pi^*_{\text{noisy}}}_{\text{noisy}}(s_0) > V^\pi_{\text{noisy}}(s_0) + \frac{1}{2} \quad \forall \pi \in \Pi_{clean}$$

*Proof.* We construct a specific MDP, the "Tightrope Walk," to prove the claim.

**MDP Definition**

Let $M = (S, A, T, R)$ be an MDP defined as follows:

- **State Space:** The state space consists of a start state $s_0$, two parallel chains of length $k$ (the "Safe Path" and the "Tightrope"), two goal states ($g_{\text{safe}}, g_{high}$), and a failure state $s_{\text{fall}}$.

$$S = \{s_0, s_{\text{fall}}, g_{\text{safe}}, g_{high}\} \cup \{u_1, \ldots, u_k\} \cup \{v_1, \ldots, v_k\}$$

  where $v_i$ are states on the Safe path and $u_i$ are states on the Tightrope.

- **Action Space:** from $s_0$ there are two actions $a_{\text{safe}}$ to enter $v_1$ or $a_{\text{rope}}$ to enter $u_1$. From each other state, there is only action $a_{fwd}$ to continue till the corresponding goal state.

- **Transitions (Noiseless):**
    - From $s_0$ there are two actions $a_{\text{safe}}$ to enter $v_1$, $T(v_1|s_0, a_{\text{safe}})$, or $a_{\text{rope}}$ to enter $u_1$ $T(u_1|s_0, a_{\text{rope}})$.
    - Both paths are deterministic chains: $T(v_{i+1}|v_i, a_{fwd}) = 1$ and $T(u_{i+1}|u_i, a_{fwd}) = 1$, terminating at $g_{\text{safe}}$ and $g_{high}$ respectively.

- **Rewards:** The reward function $R : S \to [-1, 1]$ is defined as:
    - $R(g_{high}) = 1$ (High reward for crossing the tightrope).
    - $R(g_{\text{safe}}) = 0.1$ (Low reward for the safe path).
    - $R(s_{\text{fall}}) = -1$ (Catastrophic penalty for falling).
    - $R(s) = 0$ for all other states.

**Characterizing $\Pi_{clean}$** We first identify the set of policies that can be optimal in the noiseless environment. Note that there are only two policies: $\pi_{\text{rope}}$ that takes $a_{\text{rope}}$ from $s_0$ and $\pi_{safe}$ that takes $a_{safe}$. Denote the path created from the first policy by $A$ and $B$ from the second policy.

For any discount factor $\gamma' \in (0, 1)$, the value of the only two paths from $s_0$ is:

$$V_{\gamma'}^A(s_0) = (\gamma')^{k+1} \cdot 1$$

$$V_{\gamma'}^B(s_0) = (\gamma')^{k+1} \cdot 0.1.$$

Since $(\gamma')^{k+1} > 0$, Path A strictly dominates Path B for all $\gamma' \in (0, 1)$. Thus, the set of noiseless-optimal policies contains only the "Tightrope" policy:

$$\Pi_{clean} = \{\pi_{\text{rope}}\}.$$

**The Noisy Setting** We introduce a noise model on Path A such that at every step $u_i$, the agent falls to $s_{\text{fall}}$ with probability $\epsilon$. Path B remains deterministic. We select a noise parameter $\epsilon > 0.5$ (e.g., $\epsilon = 0.9$).

We now analyze the values of the policies in this noisy environment for an arbitrary evaluation discount $\gamma \in (0, 1)$.

- **Performance of $\pi_{\text{noisy}}^*$:** The optimal policy in the noisy environment is to choose the Safe Path ($\pi_{\text{safe}}$). Since rewards are non-negative on this path:
$$V_{\text{noisy}}^{\pi_{\text{noisy}}^*}(s_0) = V_{\text{noisy}}^{\pi_{\text{safe}}}(s_0) = 0.1\gamma^{k+1} > 0$$

- **Performance of $\pi \in \Pi_{clean}$:** The only policy in $\Pi_{clean}$ is $\pi_{\text{rope}}$. Under noise $\epsilon$, the agent falls at the first step with probability $\epsilon$ (incurring $-\gamma$) or proceeds with probability $1 - \epsilon$ (receiving at most $\gamma$).
$$V_{\text{noisy}}^{\pi_{\text{rope}}}(s_0) \le \epsilon(-\gamma) + (1 - \epsilon)(\gamma) = \gamma(1 - 2\epsilon)$$

Since we chose $\epsilon > 0.5$, the term $(1 - 2\epsilon)$ is strictly negative. Thus:

$$V_{\text{noisy}}^{\pi_{\text{rope}}}(s_0) < 0$$

Since $V_{\text{noisy}}^{\pi_{\text{noisy}}^*}(s_0) > 0$ and $V_{\text{noisy}}^{\pi_{\text{rope}}}(s_0) < 0$, the noisy-optimal policy is strictly superior for **all** $\gamma \in (0, 1)$. This proves claim 1.

**Proving the Margin (claim 2 in theorem statement)** We now show the stronger condition for large $\gamma$. Let $\epsilon = 0.9$. The value difference is:
$$V_{\text{noisy}}^{\pi_{\text{noisy}}^*}(s_0) - V_{\text{noisy}}^{\pi_{\text{rope}}}(s_0) \ge 0.1\gamma^{k+1} - (-0.8\gamma)$$
$$= \gamma(0.8 + 0.1\gamma^k)$$

For sufficiently large $\gamma$ (specifically $\gamma > 0.625$), this difference strictly exceeds 0.5. Thus, for large $\gamma$, the dominance holds by a margin of at least $1/2$. $\square$

# B. Additional Grid Experiments

## B.1. Sticky Noise

In this section, we extend our experiments in Section 8.1 for reset and uniform noise to sticky noise. We reuse the same experimental setup. As $\gamma$ is varied within the interval $[0.70, 0.99]$, we compute the optimal policies on the sticky noise MDP with $\beta = 0.7$. We compare each policy with the optimal policies on the clean MDP with varying discount factors using the action distance metric described in Section 8.1, and we plot the results in Figure 7. As predicted by Theorem 4.1, we observe an exact equivalence when the clean MDP has a certain smaller discount factor than the noisy MDP.

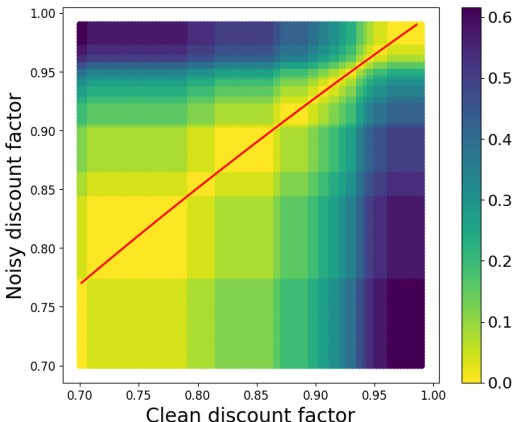

*Figure 7.* The distance between optimal policies on the clean hallway MDP and the noisy hallway MDP with uniform noise with $\beta = 0.7$. The red line depicts the $(\gamma, \gamma')$ pairs predicted by Theorem 4.1.

## B.2. Discounted and Noisy Behavior for Hallway

Recall the hallway MDP defined in Section 8.1 and depicted in Figure 2. To aid in comparing how the discount factor and noise level affect the behavior of optimal policies the hallway map, we provide a qualitative analysis of these optimal policies change as these two factors are varied.

Intuitively, policies in the Hallway map must balance between seeking higher future rewards by moving right and accepting immediate reward by moving up into the first row. We therefore see that optimal policies have a threshold in the bottom row, to the left of which they will move toward higher rewards, and to the right of which they will move upwards to collect rewards. Similarly, optimal policies also have a threshold in the top row, the left of which they will move down to find better rewards, and to the right of which they will stay where they are. As the discount factor is increased, optimal policies are more willing to seek higher reward as the time it takes to do so is less penalized, so these thresholds will shift leftward and encourage more states to go towards areas of higher reward. Note that the reward states are not terminal.

To see this behavior more clearly, we depict examples of optimal policies in Figure 8 for $\gamma = 0.90$ and $\gamma = 0.95$ on the clean MDP. For $\gamma = 0.90$, the threshold in the top row occurs at reward 3, and the threshold in the bottom row occurs in the column with reward 5. For $\gamma = 0.95$, the threshold in the top row occurs at reward 7, and the threshold in the bottom row occurs in the column with reward 9.

For the noisy MDP with uniform noise, we see a similar trend. There are again thresholds of behavior for the top and bottom rows, and these thresholds move leftward as the level of noise increases. However, the exact relationship between these two thresholds is not necessarily the same as with the clean MDP. In Figure 9, we give examples of the optimal policies for the noisy Hallway MDP with $\epsilon = 0.1$ and $\epsilon = 0.35$ for fixed discount factor $\gamma = 0.95$. We note that the optimal policy for $\gamma = 0.95$ and $\epsilon = 0.35$ on the noisy MDP is different than the optimal policy for $\gamma = 0.90$ on the clean MDP despite having the same threshold in the top row. Nonetheless, we see that noise produces optimal policies close to those induced by lower discount factors. Intuitively, this may be due to uniform noise slowing the down the agent as it tries to reach the high-reward areas. This can informally be seen as a form of sticky noise, which matches our theoretical results giving an exact equivalence between sticky noise and discounting.

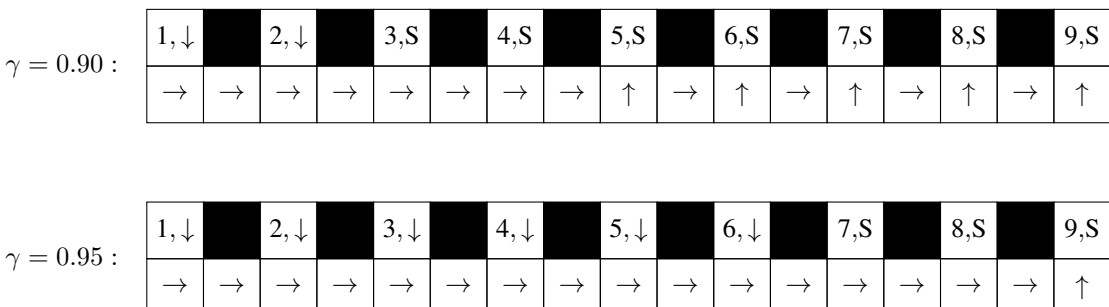

Figure 8. The optimal policy for the clean hallway MDP for $\gamma = 0.90$ and $\gamma = 0.95$ respectively. Squares are labeled with either an arrow pointing toward the direction prescribed by the optimal policy, or the letter S to mean that the optimal policy will stay still.

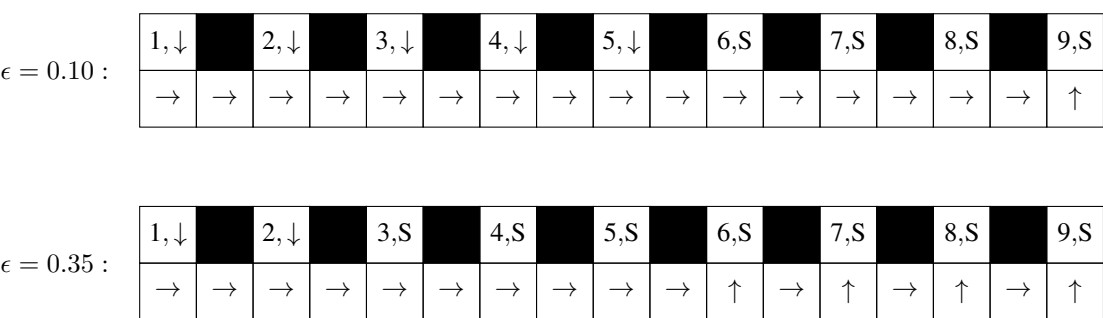

Figure 9. The optimal policy for the noisy hallway MDP with $\gamma = 0.95$ for $\epsilon = 0.1$ and $\epsilon = 0.35$ respectively.

## B.3. Non-equivalence of Noise and Discounting

While uniform noise often exhibits a quasi-equivalence with a change in the discount factor such as through Theorem 6.1, this is not always the case, even in grid environments which naturally satisfy the reversibility property. We observe that this largely occurs whenever noise can immediately cause the agent to lose substantial amounts of reward due to violations of the slowly-changing reward assumption. One example of this is given in the tightrope MDP in Section 7.2, where noise can send the agent to a large negative reward state it would not have gone to otherwise. In this section, we further explore this nuance by providing and analyzing an example of a map for which equivalence breaks down.

We examine Split, a map where we view the agent as beginning at the bottom-left corner. The only positive reward state is in the bottom-right corner with +1 reward. To get to the reward, the agent can either take a longer upper path, or take a shorter but riskier lower path that is adjacent to states with $-4$ reward. The Split MDP can be seen a variation of the Tightrope MDP where the agent is able to leave the negative reward state. The Split MDP is depicted in Figure 10.

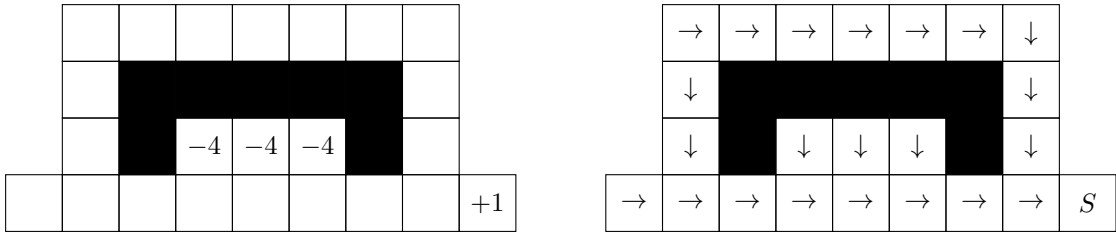

Figure 10. **The Split MDP.** On the left, we depict the Split map. On the right, we depict an optimal policy for the noiseless Split MDP for any discount factor.

The primary decision a policy must make is whether to take the safe upper path or the risky lower path. When there is no noise involved, the choice is obvious as the agent can always traverse the lower path without receiving a penalty. The optimal policy will therefore take the shortest path to the reward, as depicted in Figure 10. In contrast, when we add noise, the lower path is no longer always optimal as there is now a chance for the agent to suffer a penalty when noise activates. This causes the optimal policy to more often send the agent to the upper path instead, as we show in Figure 11.

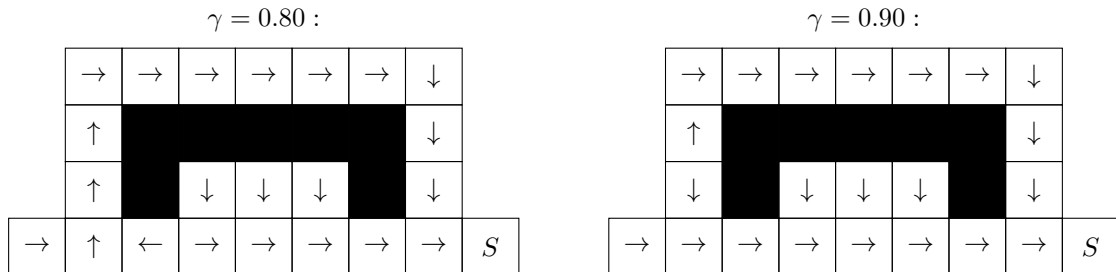

*Figure 11.* The optimal policy for the noisy Split MDP for $\epsilon = 0.3$ with $\gamma = 0.80$ and $\gamma = 0.90$ respectively.

We notice that the optimal policy in the noisy MDP behaves more similarly to the clean optimal policy for higher discount factors. In particular, when the discount factor is $0.8$, the optimal policy chooses not to go down the lower path unlike in the clean MDP. However, when the discount factor is increased to $0.9$, the optimal policy returns to heading down the lower path, albeit for one fewer starting state than before. This phenomenon occurs because, when the discount factor is higher, later time steps are factored in more than before. Since the aim of an agent in Split is to reach the positive reward goal state, the primary penalty to reward inflicted by noise occurs at earlier time steps. This means that noise in the MDP is more disruptive when the discount factor is smaller, and it is conversely less disruptive when the discount factor is large.

In general, we can consider certain states to be more dangerous if their rewards are likely to fall more greatly when visiting neighboring states. We may expect noise to align more closely with discounting whenever agents spend a greater fraction of their time in states that are less dangerous, and when this time is less heavily discounted compared to time spent in more dangerous states. This can be viewed as an informal relaxation of the slowly-changing reward assumption for when rewards are typically but not always slowly-changing in an MDP.

## C. Reacher Environment Setup

To evaluate the robustness of policy learning under continuous control constraints, we extend the classical grid-based maze task to a generalizable 2-Link Reacher environment. Developed within the `Gymnasium`(Towers et al., 2025) and `MuJoCo`(Todorov et al., 2012) frameworks, this environment reproduces the physics and reward logic of the standard `Reacher-v5` benchmark while permitting variable link configurations to test scalability and generalization.

The agent controls a planar robotic arm consisting of 2 links connected by hinge joints. The objective is to control the arm to a target position $g \in \mathbb{R}^2$ spawned stochastically within the workspace.

**Initialization**    The initialization of the `Reacher-v5` environments follows the default setup with in the document:

**Target Generation:** The target position $g = (x_g, y_g)$ is sampled uniformly from a disk of radius $r = 0.2$. Coordinates are rejected and re-sampled until satisfying $x_g^2 + y_g^2 < r^2$, guaranteeing that the target lies within the reachable workspace of the arm.

**Joint Configuration:** The initial joint positions $q \in \mathbb{R}^N$ and velocities $\dot{q} \in \mathbb{R}^N$ are initialized near the zero state with additive uniform noise to prevent deterministic starting conditions:

$$q \sim \mathcal{U}(-0.1, 0.1), \quad \dot{q} \sim \mathcal{U}(-0.005, 0.005) \tag{5}$$

**State Space**    The state space is continuous. The state vector $s_t$ provides a geometric description of the arm relative to the target:

$$s_t = [\cos(q), \sin(q), g, \dot{q}, (\mathbf{p}_{tip} - g)] \tag{6}$$

where $\cos(q), \sin(q) \in \mathbb{R}^N$ represent the joint angles, $g \in \mathbb{R}^2$ is the target position, $\dot{q} \in \mathbb{R}^N$ are the angular velocities, and $(\mathbf{p}_{tip} - g) \in \mathbb{R}^2$ is the 2D vector difference between the fingertip and target. For the standard 2-link case ($N = 2$), this results in an 10-dimensional observation space.

**Reward Structure**    We employ a dense reward function that incentivizes precision while penalizing excessive energy expenditure. A critical modification in our experimental setup, compared to standard Gymnasium benchmarks, is the amplification of the control cost weight to encourage smooth, low-energy trajectories. The reward $r_t$ at each timestep is defined as:

$$r_t = -w_{dist} \|\mathbf{p}_{tip} - g\|_2 - w_{ctrl} \|a_t\|_2^2 \tag{7}$$

where $a_t$ represents the applied torque actions. In our experiments, we set the distance weight $w_{dist} = 1.0$ and the control weight $w_{ctrl} = 5.0$. This elevated control penalty (typically $w_{ctrl} = 1.0$ in default setting) encourages the policy to take smaller actions. This is done so that the agent must balance between reaching the target quickly and keeping the penalty low, allowing the optimal behavior to change with the discount factor.

**Noise Injection Protocols**    To assess policy stability rigorously, we introduce random action noise to simulate one of the noise injection way for Reacher. With probability $p_{noise}$, the intended action $a_t$ is replaced by a random action $a_{rand} \sim \mathcal{U}(-1, 1)$ sampled from the action space. Structurally, this represents a more aggressive form of entropy injection than the sticky noise model. Where sticky noise dilutes the future by "pausing" the agent, random action noise actively diverges the agent's trajectory. This accumulation of variance effectively affect long-term predictions unreliable.

**Experiments Parameters and Optimization Techniques**    All experiments were conducted with a fixed episode length of 2,000 steps. The final policies were trained for a total of 100,000,000 timesteps to ensure convergence and maximize performance validation. Prior to this final training phase, we utilized the `Optuna` framework to identify the optimal hyperparameter configuration. This tuning process consisted of 10 trials, where each candidate model was trained for 1,000,000 steps using a Tree-structured Parzen Estimator (TPE) sampler and a Median Pruner to efficiently discard unpromising trials. The hyperparameter search space was defined as follows:

- **Learning Rate:** Sampled from a log-uniform distribution in the range $[1 \times 10^{-5}, 3 \times 10^{-4}]$.

- **Entropy Coefficient:** Sampled from a log-uniform distribution in the range $[1 \times 10^{-8}, 1 \times 10^{-2}]$.

- **Batch Size:** Selected categorically from the set $\{256, 512, 1024, 2048, 4096, 8192\}$.

We employed the Proximal Policy Optimization (PPO) algorithm implemented in `Stable-Baselines3` for all training sessions. While the parameters above were optimized per trial, we maintained fixed values for the remaining hyperparameters, including a clipping range of 0.2, 10 optimization epochs per update, and a rollout buffer length of 1024 steps.

**Computational Resources**  All the experiments are done within Intel(R) Xeon(R) CPU E5-2640 v3 @ 2.60GHz with 200GB RAM and 12 cores.

## D. Reacher Environment with Reset Noise

In addition to standard random noise, we evaluated the Reacher environment under the influence of *reset noise*. We define reset noise as a stochastic process where, at each timestep with probability $p_{\text{noise}}$, the environment spontaneously resets the system to a new initial state ($r_t = 0$).

Theoretically, Theorem 5.1 establishes that solving an MDP $M_\alpha$ subject to reset noise is equivalent to solving a standard MDP $M'$ where the discount factor is reduced from $\gamma$ to $\gamma\alpha$. However, our empirical results reveal a significant discrepancy between this theoretical equivalence and practical training stability.

In practice, training an optimal policy directly within the reset noise environment is computationally difficult and unstable due to frequent interruptions in the agent's trajectory. As illustrated in Figure 12, higher reset noise leads to policy failure; the agent fails to reach the target, collapses toward the center of the joint space, and exhibits erratic spinning behavior.

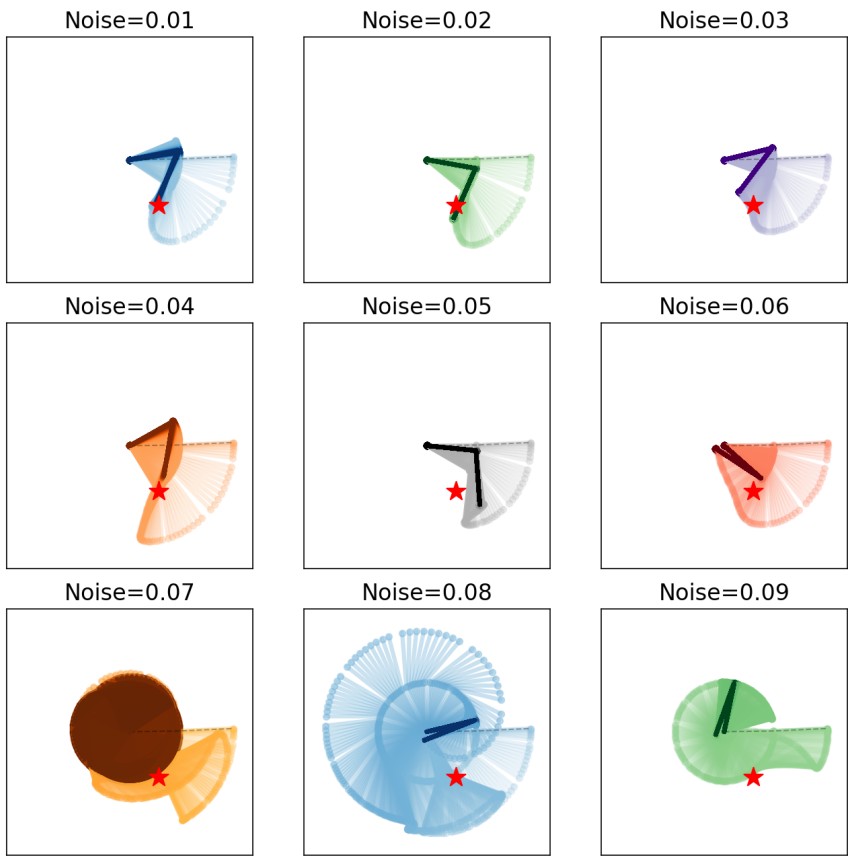

*Figure 12.* Arm Trajectory Comparison under varying reset noise levels. As noise increases beyond 0.06, the agent loses the ability to localize the target (red star) and enters a regime of high-variance, unstable motion.

Therefore, we adopted a relaxed experimental approach: instead of training directly in the noisy environment, we compared models trained on a clean environment. By leveraging the implications of Theorem 5.1, we hypothesize that the optimal policy for a noisy environment corresponds to a specific policy from the clean environment—specifically, one trained with the adjusted discount factor $\gamma\alpha$.

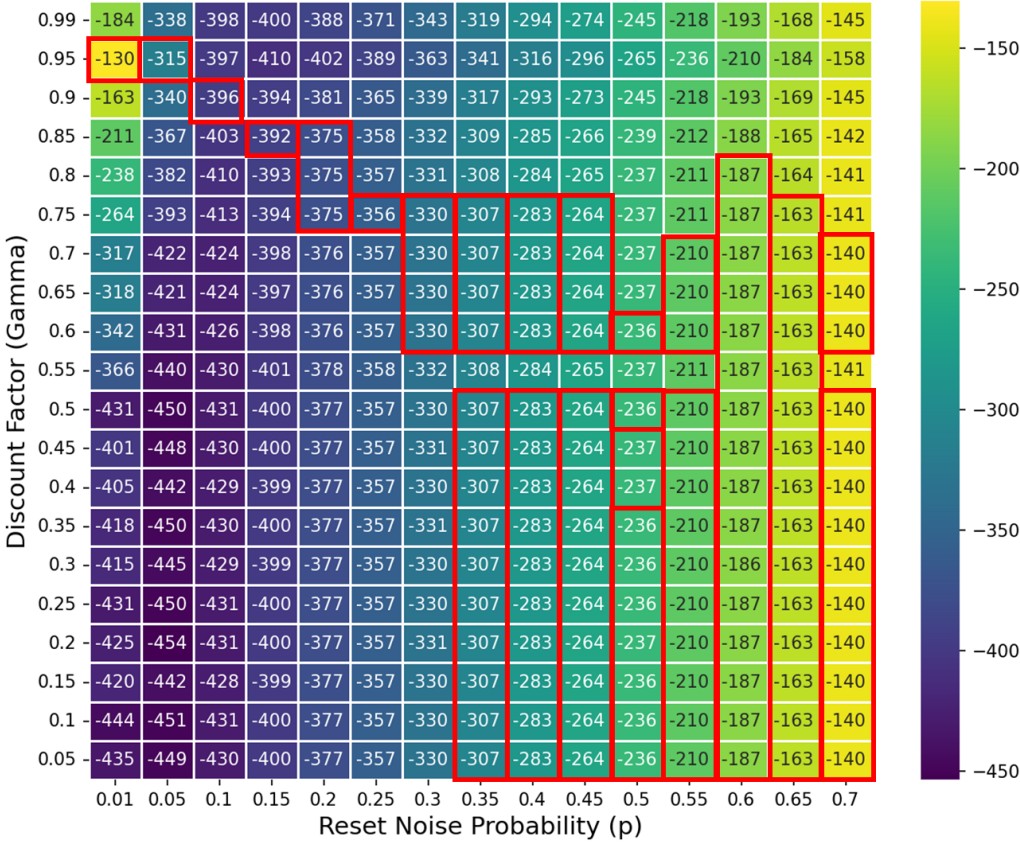

*Figure 13.* Performance Heatmap across Reset Noise Probability ($p$) and Discount Factor ($\gamma$). The red boundaries highlight the regions of optimal performance.

The heatmap in Figure 13 provides empirical evidence for Theorem 5.1. The optimal performance regions (highlighted in red) demonstrate that as the reset probability $p$ increases, the optimal discount factor $\gamma$ tends to decrease, and it is showing a diagonal trend. This confirms the theoretical prediction that reset noise acts as an implicit discount on future rewards. Specifically, the shift of the yellow-colored optimal band toward the bottom-right suggests that in high-noise regimes, the most successful policies are those that prioritize immediate rewards, aligning with the mathematically scaled discount $\gamma\alpha$.

