# OpenReview forum: "Noise as a Natural Regularizer in Markov Decision Processes: Connecting Environmental Stochasticity and Policy Simplicity"
_ICML.cc/2026/Conference — ICML 2026 regular_

### Official Review · Reviewer_42mi · 2026-02-26

**Soundness:** 3
**Presentation:** 3
**Significance:** 3
**Originality:** 3
**Overall Recommendation:** 4
**Confidence:** 1

**Summary:**

This article explores: why does aiming for short-term goals (heavily discounting future rewards) actually perform better in the real world than the textbooks claim? The writers reveal that everyday environmental chaos naturally caps how far ahead an agent can reliably look.

They back this up with hard math, proving that operating in a perfect, noise-free world with a short-term focus is mathematically identical to navigating a world hit by specific glitches:

Getting Stuck: Sometimes, the agent tries to make a move but just randomly freezes in its current spot.

Forced Restarts: Every so often, the agent gets magically warped right back to square one.

**Compliance With Llm Reviewing Policy:**

Affirmed.

**Final Justification:**

The rebuttal have addressed my concern and I will maintain my positive score.

**Key Questions For Authors:**

see weakness

**Limitations:**

yes

**Strengths And Weaknesses:**

**Strength**
The math here is very solid. The authors clearly prove that certain system glitches (like getting stuck or starting over) are exactly the same as planning for a shorter timeframe. They also use good math to show this mostly holds true for minor, fixable glitches, and they clearly state where this whole idea stops working. On the testing side, they use grid calculations to prove their exact formulas are right, and they use active simulations to show that adding more randomness generally acts just like looking less far into the future.

**Weakness**
The math for the broader "stuck" glitches is confusing and relies on strict rules that aren't well explained. Also, the proof in Section 6 assumes an impossible scenario where the agent can perfectly see and undo errors; they don't explain how a normal system could actually do this. For the active simulations, the testing is too basic—they forgot standard things like error bars and multiple runs. There is also a major mistake: they say a tracking tool measures "actions," but it actually measures "scores," which needs to be fixed. Finally, they need to build a specific test just to check if the math in Section 6 holds up in practice.

---

> ### Author Rebuttal · Authors · 2026-03-31
>
> Thanks for the review and the questions, which we address in order:
>
> **Stuck glitches:**
>
> We think that the reviewer is referring to the sticky noise section and possibly the “generalized sticky noise” results specifically. We kindly ask which parts of the section were unclear e.g. the assumptions, the proofs, the results, etc.. We are happy to clarify any confusions related to the writing, and we appreciate any suggestions on ways to improve the definitions and exposition for subsequent revisions.
>
> **Observability of noise:**
>
> The assumption that noise is perfectly observed is convenient but unnecessary. In practice, $\pi’$ need not be implemented exactly, and the agent may simply take the action with the largest expected future reward and achieve performance that is lower-bounded by our results. You can think of the proof as covering the case where attempting to return to the previous state is the best option. If the agent detects a better option based upon its Q-values, it can take that option and do at least as well.
>
> **Reversibility of noise:**
>
> This assumption holds for many planning and navigation environments. When it doesn’t, the agent may still have some form of ability to account for noise, such as by modifying future actions to gradually undo the effect of noise. At the high level, the ability to (approximately) undo the effects of noise is what allows us to relate noise to discounting: The effect of noise can be converted to a time cost, which then becomes a discounting cost. The proof adopts one particular mechanism of how this could work to make the analysis easier, but we expect the general principle will hold more broadly - and this is supported by our experiments with the Reacher environment, which demonstrate that our results hold qualitatively even when true reversibility does not hold.
>
> **Error bars & multiple runs:**
>
> The grid experiments use exact dynamic programming so there is no randomness (Figure 3). Figure 4 shows qualitative trajectory visualizations. For Figures 6 and 12, the purpose is to show when policies are alike. If we expanded our notion of similarity to include similarity within a confidence interval, then it would make more policies look similar, i.e., the red boxes would get larger but the same trend would hold. We will add more detailed plots in the appendix in subsequent revisions for completeness and transparency.
>
> **Tracking tool:**
>
> We believe that the reviewer is referring to the experiment depicted in Figure 6. We thank the reviewer for pointing out the mismatch between the caption and the actual method of comparing policies that we use. The reward distribution is used instead of the action distributions since, due to the nature of RL models, we cannot compare action distributions between policies directly since these models are unreliable at states that are rarely visited. Also, there may be spurious dissimilarities arising from actions that are equally good. JSM serves as a proxy for policy similarity and is also more informative than a simple metric like the total returns since it captures the distribution over rewards that policies induce. This is definitely something we can fix and clarify in the exposition.
>
> **Specific test for Section 6:**
>
> The bounds in section 6 are in a middle ground between the strong equivalences that exist for sticky and rest noise, and the extremely loose results that follow from the generic Jiang et al. theorems. Our bounds still have a $O(\frac{1}{1-\gamma})$ factor, so the worst case differences in value could still be quite large as noise ramps up. Also, the exact value function for non-trivial domains and for the nonstationary policy $\pi’$ is not something that is practical to compute, so we elected not to directly test the empirical tightness of the bounds. Nevertheless, we do expect that these bounds will be qualitatively informative in that they explain why the optimal policy for a smaller discount factor will tend to match the optimal policy in noisy domains when some looser form of reversibility holds. This is what we show with the Reacher experiments through qualitative analysis and quantitative metrics. For a simple domain like the Grid environment, we can demonstrate what the bounds in section 6 look like, via Monte Carlo simulation, if the reviewer(s) think it would be helpful.

---

> > ### Author Rebuttal · Reviewer_42mi · 2026-04-02
> >
> > Thanks the authors for the rebuttal. I will keep my positive score.

---

### Official Review · Reviewer_8AaW · 2026-03-10

**Soundness:** 3
**Presentation:** 3
**Significance:** 2
**Originality:** 3
**Overall Recommendation:** 4
**Confidence:** 3

**Summary:**

The authors investigate the effect of different types of structured transition noise in the Markov decision processes (MDP). They relate the effect of the noise to the horizon length that is determined by the discount factor. The authors consider the setting where the reward is a function of the state only and does not depend on the action. The paper considers four cases: sticky noise, reset noise, slowly-changing reversible MDPs and general transition noise. The authors show that the sticky and reset noise settings are equivalent to non-noisy MDPs with smaller discount factors. For the slowly-changing reversible MDP setting, the authors provide a bound for the performance loss. They also state that obtaining the bounds for the general transition noise setting is not possible. The authors also provide some experimental results for two different environments.

**Compliance With Llm Reviewing Policy:**

Affirmed.

**Final Justification:**

The rebuttal addresses my concerns. The main limitation remains as the strong assumptions on the MDP. The authors acknowledge this limitation and elaborate on the assumptions in the paper.

**Key Questions For Authors:**

1. Could you elaborate on the reversibility assumption? How restrictive is this assumption? Could you give some examples of the settings which can be modeled by this assumption?
2. Could you explain what $\pi'$ in Theorem 6.1 represents? While the proof sketch for Lemma 6.2 mentions how this is constructed, why do we care about this policy? Why do we want this specific policy to upper bound the optimal value for the discount factor $\gamma'$?
3. Could you provide some examples/discussion about how tight the bound in Theorem 6.1? Could you also provide comparisons between your bounds and the ones in Jiang et al., which were discussed in Section 7.1?
4. Theorems 4.1 and 5.1 states the equivalence of optimal policies. This implies the optimal behavior stays the same. However, the other theorems provide some equalities/inequalities for the values of these optimal policies. I am not sure if this leads to conclusions about the optimal behavior immediately. Changing the discount factor should also change the scale of the value functions. Could you discuss what does these inequalities imply about the policies? Do your results (theoretically) guarantee that the optimal policies are also close in some metric?
5. I am not quite sure what conclusion I should deduce from the results. My understanding is as follows: If I aim to act (near-)optimally in a noisy environment for some effective horizon induced by $\gamma'$, I should increase the discount factor $\gamma$ I optimize for. Then, by the results in the paper, I will behave optimally under the reduced discount factor. However, this is not quite the conclusion in the paper: "By limiting the effective planning horizon, environmental stochasticity can make simpler decision structures sufficient, offering a route to interpretability that emerges naturally rather than being enforced by design." Here, this effect is framed as if the effective horizon reduction due to the noise is something desirable. However, if I aim to optimize for a specific $\gamma'$, the noise would force me to use a larger discount factor $\gamma$, which leads to worse convergence rates. Could you discuss further actionable insights one can deduce from your results?

**Limitations:**

Yes.

**Strengths And Weaknesses:**

1. I think it is interesting and useful to understand how noise effects the dynamics. The paper also gives clear conclusions on the equivalences between the sticky and reset noise settings and discounting.
2. The section slowly-changing reversible setting is rather unclear. The reversibility assumption seems too restrictive. Theorem 6.1 is hard to interpret. I think an example that illustrates the bound given by Theorem 6.1 for varying $\gamma$ and $\epsilon$ values would be helpful. I tried to elaborate on this in the questions below.
3. The authors refer to some prior results from Jiang et al. in Section 7.1. However, they avoid providing the exact result and provide some description. I think it would be much more clear if the authors simply add the exact statement from this paper. I also believe that a direct comparison between two results would be helpful.

---

> ### Author Rebuttal · Authors · 2026-03-31
>
> We thank the reviewer for the feedback and insightful questions, which we have addressed below:
>
> **Q1.** Reversibility often holds exactly in planning and navigation environments as the agent state can be controlled precisely. However, reversibility does not always hold, particularly in cases where the agent cannot reverse the effects of noise in a single timestep or with probability 1, or when it does not have control over the entire state of the environment. In these cases, the agent may still have some ability to account for noise, by modifying future actions to gradually cancel out noise or by only approximately returning to their original state. At the high level, the ability to (approximately) undo the effects of noise is what allows us to relate noise to discounting: The effect of noise can be converted to a time cost, which then becomes a discounting cost. We examine one mechanism of how this could work to make the analysis easier, but we expect the general principle will hold more broadly - as supported by our experiments with the Reacher environment, which demonstrate that our results hold qualitatively even when reversibility does not hold exactly.
>
> **Q2.** $\pi’$ is a policy that is constructed via a small modification to the discounted policy $\pi^\ast_{\gamma’}$, and the specifics of $\pi’$ are not as important as the fact that it is easily computable from $\pi^\ast_{\gamma’}$. From a theoretical perspective, $\pi’$ is a convenient transformation of $\pi^\ast_{\gamma’}$ that performs well on the noisy environment and is only slightly more complex, connecting noise with the shorter planning horizon induced by $\gamma’$. From a practical perspective, if the user can estimate the noise distribution, then they can produce a nonstationary policy $\pi’$ by utilizing $\pi^\ast_{\gamma’}$ without the need for refitting in the noisy environment, and our results would ensure that this procedure is well-performing. The agent can also choose to instead choose the action with the largest Q-value, and our results would lower-bound the performance of such a policy with that of $\pi’$.
>
> **W3 and Q3.** Theorem 6.1 follows from Lemmas 6.2 and 6.3 as well as the assumption that noise decreases the optimal value function, each of which is individually tight. Lemma 6.2 is tighter whenever noise is chosen adversarially to decrease the reward as much as possible. For instance, we give such an example of a Grid MDP in the proof of Lemma 6.2 where the optimal policy moves in a single direction to chase dense, strictly increasing rewards, but the noise model moves the agent in the exact opposite direction. Lemma 6.3 is a looser result to bound the change in value function as the discount factor changes, and it is tighter whenever the rewards “flip” when the weightings from the two discount factors cross each other. Overall, our results are strong in the worst-case, but we would expect the quantities in question to be closer in-practice.
>
> Comparing with Jiang et al., our bounds have a dependence on $O(\frac{1}{1-\gamma})$, while theirs scale with $O(\frac{1}{(1-\gamma)^2})$. Both are provably tight, but their bounds are larger since they use weaker assumptions. If we allow arbitrary differences in transition models that are bounded only by $\delta_P$, then it is possible to come up with cases that lead to very large differences. On the other hand, if we assume that the agent has enough control to recover from noisy consequences, then we can get better bounds.
>
> **Q4.** Good question since small differences in value functions can lead to big differences in the values of policies they imply (Baird & Williams, 1993: “Tight bounds on greedy policies based on imperfect value functions”). Theorem 6.1 was structured to bound the effect of executing the optimal policy for the lower discount MDP in the noisy MDP with the original discount factor, i.e., exactly the quantity one would care about. The interpretation of theorem 4.2 is a little more subtle but it has a similar flavor: Both sides of the bound consider what happens when the optimal policy for an MDP with a smaller discount factor is executed in a noisy MDP with a larger discount factor.
>
> **Q5.** We would like to clarify the direction of our result: If the goal is to optimize over some noisy environment with discount factor $\gamma$ (not $\gamma’$), then we show that it is approximately equivalent to optimize over the clean environment with the **lower** induced discount factor $\gamma’$. In practice, this implies that noise naturally induces a shorter effective planning horizon, suggesting that it is more likely that simpler policies perform well on the noisy environment. If the practitioner has access to a cleaner version of the environment, such as through a simulator, then they can train with a smaller discount factor on the clean environment (with **faster** convergence rates) without losing much in optimality when shifting to the noisy environment.

---

> > ### Author Rebuttal · Reviewer_8AaW · 2026-04-04
> >
> > I thank the authors for their response. I increase my score.

---

### Official Review · Reviewer_r2CE · 2026-03-12

**Soundness:** 3
**Presentation:** 3
**Significance:** 2
**Originality:** 4
**Overall Recommendation:** 4
**Confidence:** 4

**Summary:**

This paper takes a noval perspective on how noise influence optimal strategy in discounted MDPs. Rather treating noises to be adversarial, as in existing robust MDP literature, this paper studies structured state transition noises and show that some noise classes influence the optimal policy by shortening the planning horizon. Specifically, the paper identifies and proves that the following three types of noises shortens the planning horizon in the optimal policy:
    - sticky noise, where there is probability not switching states
    - reset noise, where there is probability switching to states from a fixed distribution
    - reversible and slowly changing noise, where noise only leads to noisy states with rewards similar to the original state and a deterministic path back to the original state

**Compliance With Llm Reviewing Policy:**

Affirmed.

**Final Justification:**

The rebuttal addressed most of my concerns, and I maintain my original accessment towards acceptance. The main limitation is that the counter example (tightrope) only establishes impossibility results for a small class of corner cases.

**Key Questions For Authors:**

Please address my concerns and questions in the weakness section. I am willing to raise the score if the weaknesses are properly addressed.

**Limitations:**

Yes.

**Strengths And Weaknesses:**

# Strengths
- The perspective of this paper is noval and fundamentally different from other exiwsting perspectives, including robust MDPs.
- The paper successfully identifies three classes of structured noises whose influence can be characterized by an effective planning horizon.
- The theory is backup by experiments.

# Weaknesses
- Beyond characterizing the optimal or near-optimal policy, how can the results help in terms of policy learning?
For example, as a starting point, can one learn a paramterized policy that takes a noise level and outputs the action distribution for that noise level? In this case, one doesn't need to retrain the policy of the noise distribution changes. It would be great if the authors can come up with more results that facilitate policy learning in noisy environments.
- The the claim on assumptino is weak in section 7.1. The main argument relies on Theorem 2 in Jiang et al. (2016). However, this theorem only provides an upperbound of the value difference on the `noise level' (or transition kernel distance). It would be better if the authors can develop a theorem that lower bounds the value different using the transition kernel distance.
- The paper only covers a small subspace of all noises/MDPs. This paper identifies three structural assumptions on the (MDP, noise) pair where transition noises change the policy planning horizon. However, these structured classes only covers a small portion of all MDPS/noises. I strongly encourage the authors to explore (MDP, noise) classes that rule out the extreme corner cases, e.g. the tigerope example, and to character the optimal policy for a much larger space of (MDP, noises).

---

> ### Author Rebuttal · Authors · 2026-03-31
>
> Thank you very much for your review and feedback.
>
> **W1: Noise-parameterized policy learning suggestion.**
>
> We find your suggestion about learning a policy parameterized by the noise parameter quite intriguing. We’re not aware of anything like that in the literature, so we think it might be a novel approach. We are aware of some work that solves for MDPs with many different discount factors as a way of approximately covering the space of discount factors. While we think your suggestion would be an interesting complement to the topic of our paper, but we don’t think it would subsume it. Our focus is on understanding the relationship between noise and the effective horizon, and a key message of our paper is that this relationship is complex, with some noise models leading to tidy theoretical results, and others less so.
>
> **W2: Lower bound on value difference missing.**
>
> Section 7.1: Theorem 2 from Jiang et al. (please note that we are referring to the 2016 IJCAI paper here, not the 2015 AAMAS paper) bounds exactly the quantity we study in our paper and exactly the quantity a practitioner would care about: The loss expected from using the optimal policy from an MDP with a smaller discount on a different MDP with a larger discount. $\delta_P$ is the transition kernel distance. This can be interpreted as the 1-norm distance between the original MDP and the noisy MDP. As for the tightness of their bound, Jiang et al. provide examples for which their bound holds exactly, and in general a nontrivial lower bound cannot be given since neither the optimal policy nor their value functions necessarily change under perturbations to the transition function. For instance, if noise is added that promotes the previously optimal action, then the optimal policy may not change at all, even for high levels of noise. Likewise, the optimal policy could change in a way that compensates for noise, leading to no net change in the value function. We therefore focused on the asymptotics of their upper bound compared to ours in Section 7.1.
>
> **W3: Coverage of noise/MDP space too narrow**
>
> The three noise models we study (sticky, reset, and slowly-changing reversible) are not arbitrary. They are practically motivated and collectively span a range from exact equivalence to approximate optimality guarantees. We share your interest in discovering useful properties of more general noise models, and the tightrope counterexample shows that a fully general characterization is impossible, which justifies why structured assumptions are necessary in the first place. Since characterizing the full space of (MDP, noise) pairs is not a tractable goal, we believe that identifying structures that can be theoretically characterized is a meaningful contribution. We agree that extending this to additional practically relevant noise models, especially noise models befitting continuous domains, is a worthwhile direction for future work.

---

> > ### Author Rebuttal · Reviewer_r2CE · 2026-04-03
> >
> > I thank the authors for their rebuttal, and I will maintain my positive score towards acceptance. On W3, I still encourage the authors to establish conditions so that extreme examples including the tightrope are ruled out and a general characterization is possible.

---

### Official Review · Reviewer_7ASi · 2026-03-12

**Soundness:** 3
**Presentation:** 3
**Significance:** 2
**Originality:** 3
**Overall Recommendation:** 5
**Confidence:** 3

**Summary:**

This paper studies the effect of noisy MDPs on the optimal policy, and the relationship between the discount factors of the policy of noisy MDP and that of noise-free MDP.

**Compliance With Llm Reviewing Policy:**

Affirmed.

**Final Justification:**

I gave my final thought in my previous feedback, score and rebuttal.

**Key Questions For Authors:**

What are the limitations of the considered noise models in evaluating more complex continuous environment?

There are also some theorems on the effect of noise models on the optimal value-function. Cannot it be empirically evaluated based on some experiments?

**Limitations:**

Some limitations of this work are well mentioned.

**Strengths And Weaknesses:**

**Strengths:**
The paper is theoretically supported by solid and interesting theorems.

The theories of the paper is validated by two experiments.

**Weaknesses:**
Although there are results on reset and uniform noise, the reviewer also expects an empirical evaluation of the sticky noise on the discount-factor and resulting policy, or explanation why this type of model is not considered.

---

> ### Author Rebuttal · Authors · 2026-03-31
>
> Thank you very much for your review and feedback.
>
>
> **W1: Empirical validation of sticky noise.**
>
>
> In the cases where there is a strong equivalence between the noisy case and smaller discount factor (such as for sticky noise), we made a conscious decision not to include experiments in the submission because they would simply validate that the math is correct. We did, however, run experiments on simpler MDPs to validate our theoretical results for ourselves, and we can add them to the appendix in a subsequent revision if the reviewers believe it would be helpful. We thought it would be more interesting to highlight experiments where the theory and experiments aren’t perfectly aligned since this is often what happens in practical reinforcement learning: Theory can guide and inform practice, but it rarely aligns perfectly with it.
>
> **Q1.Limitations of the considered noise models in more complex continuous environment**
>
> In continuous environments, sticky and reset noise remain reasonable models of noise. Slowly changing, reversible noise as a model has intuitive appeal in scenarios where the agent is trying to follow a nominal trajectory through state space, is occasionally thrown off, but has enough control to eventually get back on the nominal trajectory. This intuition carries over to the continuous case as well, but some details of the analysis are less natural in that case. For instance, under random Gaussian noise that perturbs a given action, randomness comes not from whether noise occurs but instead the strength of the perturbation. In these cases, such models of noise can be intuitively approximated by slowly-changing reversible noise by supposing that actions with small draws of perturbations are essentially noiseless, and that larger perturbations are folded into the noise model. Furthermore, exact reversibility may not occur since an agent cannot reverse the effects of noise exactly (since in continuous space it may be impossible to exactly visit a specific continuous point), but they may still return to a state close to their original state with high probability. We leave further exploration of continuous noise models to future work.
>
> **Q2. Effect of noise models on the optimal value-function**
>
> Regarding the optimal value function and experiments in the case where there is not an exact equivalence (neither sticky nor reset noise):  First, we’d like to point out that the optimal value function is less important than the optimal policy. Even in the case of sticky noise and reset noise, our results are expressed in terms of value functions but they are really about policies: They show that the optimal policy in the reduced-discount MDP is the same as the optimal policy in the noisy MDP, but the value functions won’t necessarily be the same. For the small, discrete Grid environment, we have performed experiments comparing optimal value functions, but chose to instead include our experiments comparing optimal actions as we found them more insightful. In large or continuous domains, the optimal value function and the optimal policy are infeasible to compute, so our results use a similarity metric (JSD) comparing the behavior of the best policies found in different scenarios.

---

> > ### Author Rebuttal · Reviewer_7ASi · 2026-04-01
> >
> > All of my concerns are added.
> > I also suggest the authors to add the results of sticky noise if the paper will be accepted.
> > More noise models can be considered as a future work and one of the limitation of this work is that it is considering some specific classes of noise.
> > I am also changing the score.

---

### Decision · Program_Chairs · 2026-04-30

**Decision:**

Accept (regular)

**Comment:**

This paper studies properties of MDPs under which the optimal policy is "simple", in particular making connections between noisiness in the environment and effective horizon for planning. The reviewers agree that this is an interesting and novel perspective and uniformly lean towards acceptance.